# CSV-Occ: Fusing Multi-frame Alignment for Occupancy Prediction with Temporal Cross State Space Model and Central Voting Mechanism

Ziming Zhu[1]  Yu Zhu[1]  Jiahao Chen[1]  Xiaofeng Ling[1 2]  Huanlei Chen[3]  Lihua Sun[1]

## Abstract

Recently, image-based 3D semantic occupancy prediction has become a hot topic in 3D scene understanding for autonomous driving. Compared with the bounding box form of 3D object detection, the ability to describe the fine-grained contours of any obstacles in the scene is the key insight of voxel occupancy representation, which facilitates subsequent tasks of autonomous driving. In this work, we propose CSV-Occ to address the following two challenges: (1) Existing methods fuse temporal information based on the attention mechanism, but are limited by high complexity. We extend the state space model to support multi-input sequence interaction and conduct temporal modeling in a cascaded architecture, thereby reducing the computational complexity from quadratic to linear. (2) Existing methods are limited by semantic ambiguity, resulting in the centers of foreground objects often being predicted as empty voxels. We enable the model to explicitly vote for the instance center to which the voxels belong and spontaneously learn to utilize the other voxel features of the same instance to update the semantics of the internal vacancies of the objects from coarse to fine. Experiments on the Occ3D-nuScenes dataset show that our method achieves state-of-the-art in camera-based 3D semantic occupancy prediction and also performs well on lidar point cloud semantic segmentation on the nuScenes dataset. Code will be available at https://github.com/ZeaZoM/CSV-Occ.

[1]School of Information Science and Engineering, East China University of Science and Technology, Shanghai, China [2]Shanghai Key Laboratory of Intelligent Sensing and Detection Technology, East China University of Science and Technology, Shanghai, China [3]Shanghai Motor Vehicle Inspection Certification & Tech Innovation Center Co., Ltd., Shanghai, China. Correspondence to: Yu Zhu <zhuyu@ecust.edu.cn>.

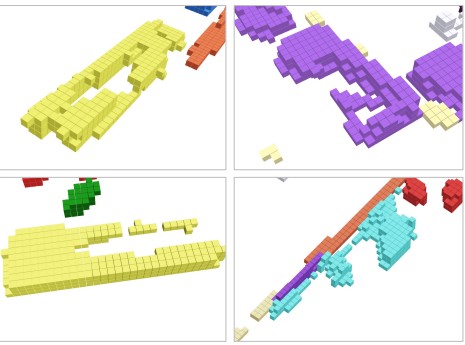

*Figure 1.* **Examples of Internal Occupancy Vacancy (IOV).** The yellow grids are the occupancy predictions of the model for "bus", purple for "truck", cyan for "construction vehicle", red for "pedestrian", orange for "barrier", light yellow for "traffic cone", and green for "vegetation". In these four examples, IOV occurred for "bus", "truck", and "construction vehicle".

## 1. Introduction

Image-based 3D semantic occupancy prediction (Mescheder et al., 2019; Peng et al., 2020) represents the structuring of 3D scenes into grids and determining whether the grids are occupied and the specific semantic categories of the occupied grids only from visual images. In the environmental understanding of autonomous driving, it provides more geometric details than 3D object detection and complements LiDAR-based perception.

Temporal information is extremely crucial for the visual system to understand the surrounding environment. For example, detecting the multi-angle geometry of highly occluded objects or inferring the relative distance of objects is very difficult for 2D images without depth information. And temporal cues can make up for the inherent deficiencies of static 2D images in these aspects. Taking the inspiration of Vision Transformers (ViTs) (Dosovitskiy et al., 2021; Liu et al., 2021; Touvron et al., 2021; Zhang et al., 2023a) as the starting point, some existing studies, such as (Koh et al., 2023; Tong et al., 2023; Lu et al., 2023; Liu et al., 2024a), will align historical BEV features (Li et al., 2022; Liu et al., 2023a) based on the attention mechanism (Vaswani, 2017)

to effectively represent the current environment. Although this brings strong learning ability, with the increase in the number of tokens, its quadratic complexity will introduce a large amount of computational overhead in downstream tasks involving large spatial resolutions. To address this challenge, inspired by VMamba (Liu et al., 2024b), Vision Mamba (Zhu et al., 2024), and Cross Attention mechanisms (Gheini et al., 2021; Lin et al., 2022) in computer vision, we propose the Cross State Space Module, a feature fusion method that extends the State Space Model (SSM) (Gu & Dao, 2023; Mehta et al., 2023; Wang et al., 2023a) to support the interaction of multiple input sequences, thereby reducing the computational complexity from quadratic to linear.

In addition, after observing the data, as shown in Fig. 1, we found that the centers of foreground objects (such as car, bus, and truck, etc.) are often predicted as empty voxels, especially for large objects. We call this problem "Internal Occupancy Vacancy (IOV)". Different from the common back occlusion problem, as shown in Fig. 2, when we increase the number of input frames, although the model successfully predicts the occupancy of the occluded surface of the target through the increase of the viewing angle, it still cannot solve the IOV. The center feature (Yin et al., 2021; Chen et al., 2023; Bai et al., 2022) is the best representation of the entire instance for 3D object detection (Lang et al., 2019; Shi et al., 2020; Wang et al., 2022; Li et al., 2022; Yang et al., 2020). However, 3D semantic occupancy prediction is different from object detection. Existing advanced 3D semantic occupancy prediction methods (Wang et al., 2023b; Li et al., 2023b; Huang et al., 2023; Li et al., 2023c; Pan et al., 2024; Wang et al., 2024; Liu et al., 2024a; Lu et al., 2023; An et al., 2024; Shi et al., 2024; Jang et al., 2024; Ye et al., 2024) perform voxel-wise classification on the 3D spatial volume feature map, while ignoring the fact that foreground voxels actually have instance aggregativity (multiple voxels belonging to the same instance have potential complementary features). To address this issue, we propose the Voting-based Enhancement Mechanism, which spontaneously learns to utilize the other voxel features of the same instance to update the semantics of the internal vacancies of the object from coarse to fine, thereby reducing the IOV.

Our contributions are summarized as follows:

- In CSV-Occ, we propose the Cross State Space Module (Cross SSM). It extends the SSM to a feature fusion method that supports multi-sequence input and uses a cascading approach to achieve the interaction of historical temporal volume features, which can enhance the model's ability to integrate temporal information.

- We also propose the Voting-based Enhancement Mechanism. This enables the model to adaptively infer the

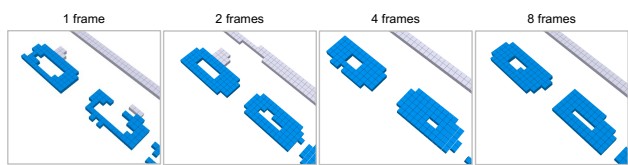

*Figure 2.* **The influence of multi-frame prediction on the IOV problem.** The blue grids are the horizontal sections of the model's occupancy predictions for two "cars", and the gray grids are "man-made". When the number of frames increases from 4 to 8, the IOV may still become more serious.

instance to which the voxel belongs and enhance the associative ability of the internal semantics of the object, enabling it to output more accurate 3D semantic occupancy representations.

- Experiments on the Occ3D-nuScenes dataset show that our method achieves the state-of-the-art performance of camera-based 3D semantic occupancy prediction. After further projecting the 3D semantic occupancy results to the lidar point cloud semantic segmentation, it also performs well on the nuScenes dataset.

## 2. Related Work

### 2.1. Temporal Modeling in 3D Perception

Camera-based 3D perception has attracted much attention in the field of autonomous driving due to its cost-effectiveness and rich visual attributes. Utilizing temporal information to compensate for the lack of depth information in single-frame 2D planar images can significantly improve the performance of camera-based 3D perception. Recently, 3D perception methods based on multi-frame images have unified this issue into the problem of voxel feature map transformation from historical frames to the current frame. One type of method (Park et al., 2022; Li et al., 2023a; Ma et al., 2024; Wang et al., 2024; Zhou et al., 2024; Liu et al., 2023b) inherits the inspiration from BEVDet4D (Huang & Huang, 2022), independently extracts and constructs the BEV feature map of each frame, and predicts the relative motion of the target between adjacent frames, thereby aligning the target-by-target features of the historical frame to the current frame position, and then concatenating with the BEV feature map of the current frame. Another type of method (Koh et al., 2023; Tong et al., 2023; Lu et al., 2023; Liu et al., 2024a) avoids explicitly predicting the relative displacement between frames. They follow the TSA paradigm proposed by BEVFormer (Li et al., 2022), predefine a learnable BEV query in the current frame, and extract information from the historical BEV feature map through the deformable attention mechanism (Zhu et al., 2020). Although this method can effectively

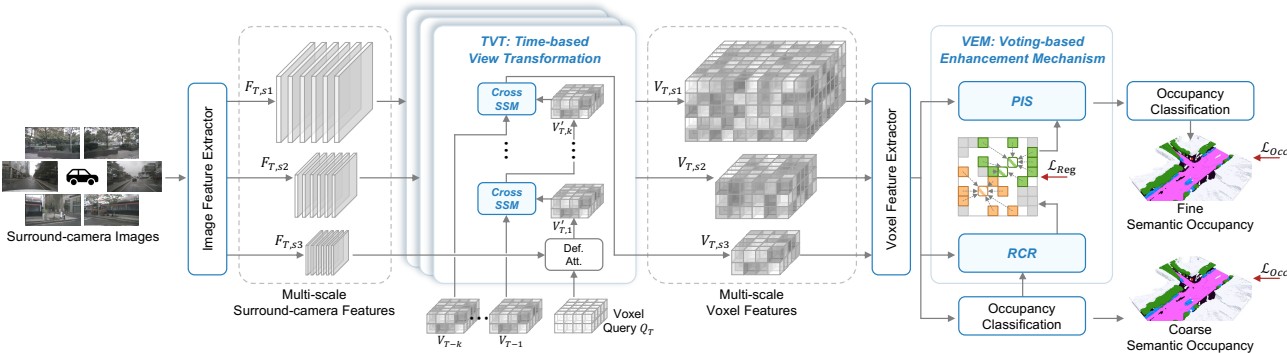

*Figure 3.* **The overall architecture of the proposed method.** First, we extract multi-scale features of multi-camera images using the image backbone. Next, we employ time-based view transformation to integrate information from multiple cameras and frames, and use the voxel backbone to upsample and concatenate 3D volume features in a multi-scale manner. Finally, the voting-based enhancement mechanism is applied to refine the coarse semantic occupancy prediction.

fuse historical information into the current frame, with the increase in the number of tokens, its quadratic complexity will introduce a large amount of computational overhead in downstream tasks involving large spatial resolutions. To address this limitation, we extend SSM to multi-sequence input for temporal information encoding, thereby achieving linear computational complexity.

## 2.2. 3D Semantic Occupancy Prediction

Recently, camera-based 3D semantic occupancy prediction has received considerable attention. This prediction aims to reconstruct the 3D scene structure from images and predict the occupancy and semantic attributes of all voxels. MonoScene (Cao & De Charette, 2022) adopts a camera-based method and 3D UNet (Ronneberger et al., 2015; Çiçek et al., 2016) architecture; TPVFormer (Huang et al., 2023) uses three perspective views to represent the 3D scene for predicting 3D occupancy; OccFormer (Zhang et al., 2023b) decomposes 3D processing into local and global transformer paths; SurroundOcc (Wei et al., 2023) achieves fine-grained results through multi-scale supervision; FB-OCC (Li et al., 2023c) uses forward-backward projection to aggregate multi-image information; PanoOcc (Wang et al., 2024) unifies 3D semantic occupancy prediction and 3D panoramic segmentation tasks; SparseOcc (Liu et al., 2024a) employs mask-guided sparse sampling to predict semantic occupancy from 3D sparse representations; OctreeOcc (Lu et al., 2023) utilizes variable granularity octree representations to adapt to object shapes and semantic regions of different sizes and complexities. To promote occupancy representation learning, we propose a new framework named CSV-Occ, which utilizes the temporal state space model and voting mechanism to optimize multi-frame fusion and update the internal features of objects.

## 3. Method

### 3.1. Problem Definition

The camera-based 3D semantic occupancy prediction task is designed to predict the dense semantic voxel volume surrounding the ego-vehicle through the analysis of multi-view images. Specifically, we leverage the multi-view image set of the current frame, denoted as $Img = \{I_1, I_2, \ldots, I_n\}$, as the model input, where n represents the index of the camera image viewpoint. The camera-based 3D semantic occupancy prediction task is formulated as follows:

$$\hat{O} = \mathcal{F}(Img) = \mathcal{F}(\{I_1, I_2, \ldots, I_n\}) \quad (1)$$

Where $\mathcal{F}$ represents the model integrating multi-view camera image information for 3D semantic occupancy prediction. The model then generates the semantic voxel volume $\hat{O} \in \{p_e, p_1, \ldots, p_C\}^{H \times W \times L}$ as the output. In this context, the parameter $p$ ranges from 0 to 1, presenting the probability of voxel grid occupancy. The variable $C$ denotes the total count of semantic classes within the scene, with $p_e$ indicating the probability that the grid contains an empty occupied voxel. Additionally, $H$, $W$, and $L$ represent the height, width, and length of the voxel volume, respectively.

### 3.2. Overall Architecture

Fig. 3 illustrates the overall architecture of our method. Given a set of surround-camera images, we first use an image feature extractor (such as ResNet-101 (Dai et al., 2017)) to extract multi-scale surround-camera features $F = \{\{f_i^j\}_{i=1}^N\}_{j=s1}^{sm}$ at $m$ scales from $N$ cameras individually. In the Time-based View Transformation (TVT) module, for the surround-camera features at each level, we utilize voxel queries to learn volume features and use the cascaded Cross

State Space Module (Cross SSM) to fuse the volume features of historical frames into the current frame to enrich the feature information. Then, the Voxel Feature Extractor is used to upsample and concatenate the multi-scale volume features to form high-resolution volume features. Next, the occupancy classification head predicts the coarse semantic occupancy based on the high-resolution volume features. In the Voting-based Enhancement Mechanism (VEM), the model explicitly predicts the relative displacement of each foreground voxel to the instance center it belongs to through Relative Central Regression (RCR), and then updates the high-resolution voxel features through the Parallel Interaction Strategy (PIS). Finally, the occupancy classification head predicts the refined semantic occupancy.

### 3.3. Time-based View Transformation

We project the reference voxels to the camera views and use deformable attention (Zhu et al., 2020) to query pixels and aggregate information (More details are in Sec. A of the Appendix).

To effectively obtain better multi-frame voxel features, we design a cascading structure after the deformable attention mechanism to gradually fuse the information in the multi-frame voxel features, as shown in Fig. 3. Based on the observation that directly adding or concatenating voxel features from different frames leads to performance degradation (as shown in the ablation experiments in Table 3), we decompose this interaction from the current frame voxel feature to the fused voxel feature $V_T \in \mathbb{R}^{h \times w \times l \times d_{\text{voxel}}}$ into $k$ cascading steps. Between the input current frame voxel feature and the fused cascaded voxel feature, we refer to the intermediate voxel feature containing information from different frames as $V'_{T,i} \in \mathbb{R}^{h \times w \times l \times d_{\text{voxel}}}$. Step by step, the model gradually fuses the information of historical frames to effectively and efficiently learn the final occupancy descriptor $V_T$.

#### 3.3.1. Cross State Space Module

Inspired by Mamba (Gu & Dao, 2023), we achieve the fusion and interaction of inter-frame 3D voxel features through SSM (Gu & Dao, 2023; Mehta et al., 2023; Wang et al., 2023a). To achieve this goal, we need to expand the vanilla Mamba into multiple inputs.

The original Attention formula (Dosovitskiy et al., 2021; Vaswani, 2017) can be approximated as a SSM through simplification, addition of new terms, and transformation (derivations are in Sec. B of the Appendix). This makes it possible for SSM to achieve the interaction of different input sequences. And we found in the derivation that the role of the $C$ variable in the state space model is similar to the query tensor in Attention (in Cross Attention (Lin et al., 2022; Gheini et al., 2021), query tensor can be obtained by mapping another input feature sequence tensor). Therefore,

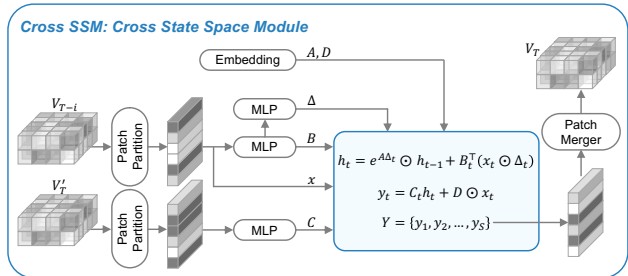

*Figure 4.* **Illustration of cross state space module.** We take the 3D feature maps of the current frame and the past frames simultaneously as the input. Where $\odot$ represents the element-wise multiplication of tensors.

we believe that the C variable in the state space model can also be obtained by mapping another input feature sequence tensor similarly.

Thus, as shown in Fig. 4, we take the 3D voxel features of the current frame $T$ and the 3D voxel features of the historical frame $T - i$ simultaneously as the input of the Cross SSM. $V_{T-i}$ is taken as $x \in \mathbb{R}^{S \times d_{\text{voxel}}}$, $B \in \mathbb{R}^{S \times d_{\text{state}}}$ is obtained by the MLP mapping of $x$, $\Delta \in \mathbb{R}^{S \times d_{\text{voxel}}}$ is obtained by the MLP mapping of $B$. $C \in \mathbb{R}^{S \times d_{\text{state}}}$ is obtained by the MLP mapping of $V'_T$. Both $A \in \mathbb{R}^{d_{\text{state}} \times d_{\text{voxel}}}$ and $D \in \mathbb{R}^{d_{\text{voxel}}}$ are learnable tensors. Among them, $d_{\text{state}}$ represents the SSM state dimension. Where S represents the length of the input sequence. Specifically, we expand $V_{T-i}$ and $V'_T$ into sequences of the same size $\mathbb{R}^{S \times d_{\text{voxel}}}$ through patch partition respectively, and then use them as the input of the state space model. Finally, the output of the state space model is reconstructed as $V_T \in \mathbb{R}^{h \times w \times l \times d_{\text{voxel}}}$ through patch merger. In the patch partition, we use simple pooling downsampling and tensor flattening without introducing any complex processes, and the patch merger is implemented by simple reshape and interpolation upsampling.

### 3.4. Voting-based Enhancement Mechanism

Based on the observation that the IOV problem is prevalent in semantic occupancy prediction (as shown in Fig. 1 and Fig. 2), we developed the RCR and PIS mechanisms, which can learn to spontaneously utilize the coarse semantic occupancy predictions and high-resolution volume features to fill the vacant semantic features inside the objects.

#### 3.4.1. Relative Central Regression

The foreground class objects often have the IOV problem (there are a total of 10 types of foreground objects such as cars, trucks, bicycles, etc. in the nuScenes dataset). In the nuScenes dataset, we only focus on the refinement of the internal voxels of these 10 types of objects and do not need

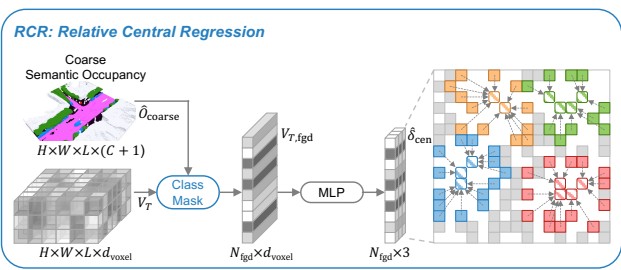

*Figure 5.* **Illustration of relative central regression.** We use the BEV plane for a simple 2D illustration, while it is actually a 3D process. Different colors of the blocks represent different categories of voxels. Gray indicates other non-foreground categories, blank indicates non-occupied areas, and the dashed arrows indicate the relative direction and distance of the central voxels voted by the voting voxels.

to refine the entire occupancy space, otherwise, it will bring a huge amount of computational redundancy. As shown in Fig. 5, we first filter out the voxel coordinates occupied by the above 10 types of objects through coarse semantic occupancy predictions, and use them as the class mask to filter the high-resolution volume features, that is, only the voxel features $V_{T,\text{fgd}} \in \mathbb{R}^{N_{\text{fgd}} \times d_{\text{voxel}}}$ of the above 10 types of objects are retained:

$$V_{T,\text{fgd}} = \text{Mask}\left(V_T, \hat{O}_{\text{coarse}}\right) \tag{2}$$

Where $\hat{O}_{\text{coarse}} \in \{p_e, p_1, \ldots, p_C\}^{H \times W \times L}$ represents the coarse semantic occupancy prediction. For these features, we predict the offsets $\hat{\delta}_{\text{cen}} \in \mathbb{R}^{N_{\text{fgd}} \times 3}$ of these voxels from the central voxels of their respective objects (including the voxel distances $\Delta x_i, \Delta y_i, \Delta z_i$ in the $H$, $W$, and $L$ directions) through the MLP:

$$\hat{\delta}_{\text{cen}} = \text{MLP}\left(V_{T,\text{fgd}}\right) \tag{3}$$

Thus, we obtain the voting relationship (that is, the voxel-relative central voxel pair) of each voxel to the central voxel of its respective object.

### 3.4.2. PARALLEL INTERACTION STRATEGY

The voxel-relative central voxel pairs can be one-to-one or many-to-one (that is, it is possible that multiple voxels vote to the same center voxel of the belonging object), as shown on the left side of Fig. 6. For each voted central voxel feature $f_{\text{cen}}^i$, there are $G_i$ voting voxels features $\{f_{\text{vot}}^{i,j}\}_{j=1}^{G_i}$ that have a voting relationship with it, where $i$ represents the index of the voxel-relative central voxel group. We hope that they can update their own features through sufficient interaction.

As shown on the right side of Fig. 6, taking $i = 1$ as an example, first, $f_{\text{cen}}^1$ is used as the key and value, and $f_{\text{vot}}^{1,j}$ is used as the query to perform cross attention calculation respectively and is added with the residual of $f_{\text{vot}}^{1,j}$ to obtain the updated feature $\dot{f}_{\text{vot}}^{1,j}$:

$$\dot{f}_{\text{vot}}^{1,j} = \text{CrossAttn}\left(f_{\text{vot}}^{1,j}, f_{\text{cen}}^1\right) + f_{\text{vot}}^{1,j}, for \ j = 1, \ldots, G_1 \tag{4}$$

For each $\dot{f}_{\text{vot}}^{1,j} \in \mathbb{R}^{1 \times D_{\text{voxel}}}$, we perform global pooling to obtain a feature descriptor $\in \mathbb{R}^1$, and then sort these feature descriptors. Use this order as the sequence of each $\dot{f}_{\text{vot}}^{1,j}$ to obtain $\widetilde{f_{\text{vot}}^{1,j}}$:

$$\left\{\widetilde{f_{\text{vot}}^{1,j}}\right\}_{j=1}^{G_1} = \text{Sort}\left(\left\{\dot{f}_{\text{vot}}^{1,j}\right\}_{j=1}^{G_1}, \text{GP}\left(\left\{\dot{f}_{\text{vot}}^{1,j}\right\}_{j=1}^{G_1}\right)\right) \tag{5}$$

Where, GP represents the global pooling function. After determining the order of $\widetilde{f_{\text{vot}}^{1,j}}$, use it as the key and value, and $f_{\text{cen}}^1$ as the query, and perform cross attention calculation in sequence to obtain the updated feature $\dot{f}_{\text{cen}}^1$:

$$\dot{f}_{\text{cen}}^1 = \text{CrossAttn}\left(f_{\text{cen}}^1, \widetilde{f_{\text{vot}}^{1,j}}\right), for \ j = 1, \ldots, G_1 \tag{6}$$

### 3.5. Supervision Signal

**Occupancy Classification.** We adopt a simple two-layer MLP for voxel-wise multi-classification (including occupancy status and semantic category) of high-resolution voxel features without other complex designs. Then, the occupancy prediction $\hat{O}_{geo} \in \{p_e\}^{H \times W \times L}$ and the semantic prediction $\hat{O}_{sem} \in \{p_1, p_2, \ldots, p_C\}^{H \times W \times L}$ are output.

**Loss Function.** The total loss $\mathcal{L}$ consists of two parts:

$$\mathcal{L} = \mathcal{L}_{Occ} + \mathcal{L}_{RCR} \tag{7}$$

The occupancy classification head is supervised by $\mathcal{L}_{Occ}$. Specifically, $\mathcal{L}_{Occ}$ consists of the focal loss ($\mathcal{L}_{focal}$) (Ross & Dollár, 2017), the Lovasz loss ($\mathcal{L}_{lov}$) (Berman et al., 2018), and the scene-class affinity loss ($\mathcal{L}_{aff}$) (Cao & De Charette, 2022). Among them, $\mathcal{L}_{aff}$ calculates the losses $\mathcal{L}_{aff}(\hat{O}_{sem}, O_{sem})$ and $\mathcal{L}_{aff}(\hat{O}_{geo}, O_{geo})$ for the semantic labels (including voxel semantic categories) and the geometric labels (including only the voxel occupancy status), respectively. $\mathcal{L}_{Occ}$ is calculated as follows:

$$\begin{aligned} \mathcal{L}_{Occ} &= \lambda_{focal}\mathcal{L}_{focal} + \lambda_{lov}\mathcal{L}_{lov} + \lambda_{aff}\mathcal{L}_{aff} \\ \mathcal{L}_{aff} &= \mathcal{L}_{aff}(\hat{O}_{sem}, O_{sem}) + \mathcal{L}_{aff}(\hat{O}_{geo}, O_{geo}) \end{aligned} \tag{8}$$

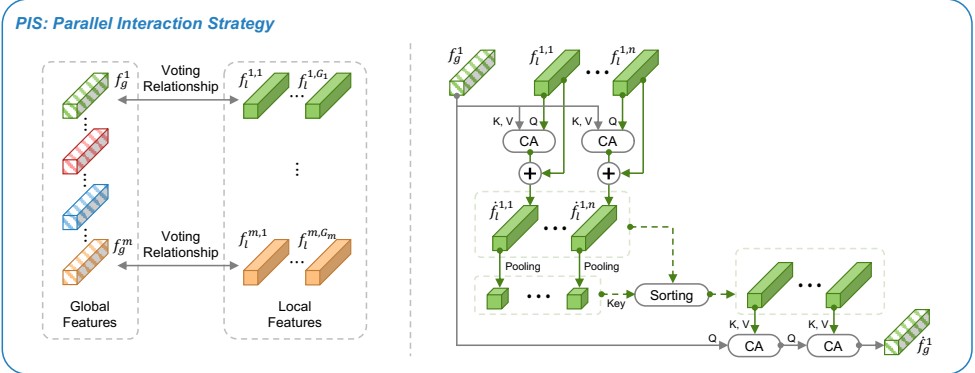

Figure 6. **Illustration of parallel interaction strategy.** The voxel-relative central voxel pairs conduct feature interaction through cross-attention calculation, thereby updating all foreground voxel features.

RCR is supervised by $\mathcal{L}_{RCR}$. Specifically, we utilize the bounding box labels, project them onto the volumetric coordinate system, obtain the voxels to which the center of each object instance belongs, and calculate the distance deviation of all voxels within the bounding box of each instance to the central voxel to obtain $\delta_{\text{cen}}$. We predict a three-dimensional vector for each voxel belonging to the foreground classes to regress the relative center offset and use the MSE loss for supervision:

$$\mathcal{L}_{RCR} = \lambda_{RCR}\text{MSE}\left(\hat{\delta}_{\text{cen}}, \delta_{\text{cen}}\right) \quad (9)$$

## 4. Experiments

### 4.1. Implementation Details

**Network.** For CSV-Occ-B, the input image resolution is $1600 \times 896$. We employ ResNet101-DCN (Dai et al., 2017) as the image backbone. The Feature Pyramid Network (FPN) (Lin et al., 2017) extracts multi-scale features with downsampling strides of 8, 16, 32, and 64. The time-based view transformation module fuses 4 frames with a frame interval of 2. The voxel feature extractor adopts 3 layers of 3D deconvolution to upsample the BEV size by $2\times$ and the height by $4\times$. The occupancy classification head consists of two layers of 128-dimensional MLP and softmax activation, generating an occupancy space resolution of $200 \times 200 \times 16$. For CSV-Occ-S, the input image size is reduced to $800 \times 448$, and we employ ResNet50 (He et al., 2016) as the image backbone, with the remaining network details the same as CSV-Occ-B.

**Training.** For CSV-Occ-B and CSV-Occ-S, we train the models on 4 NVIDIA A40 48G GPUs with a batch size set to 4. During training, we utilize the AdamW (Loshchilov, 2017) optimizer with an initial learning rate of $2 \times 10^{-4}$, weight decay of 0.05, and apply a cosine annealing strategy.

We train our models for all experiments over 24 epochs. We apply common data augmentation strategies, including color transformations, flips, rotations, and scaling in both image and 3D space.

### 4.2. Main Results

**3D Semantic Occupancy Prediction.** In Table 1, we present a comparative experiment with other state-of-the-art 3D semantic occupancy prediction methods on Occ3D-nuScenes (Tian et al., 2024) validation set. Due to the default image backbone of FB-OCC (Li et al., 2023c) being R50 (ResNet50) (He et al., 2016) and the resolution of the input images differing from that of other methods, we modified the image backbone and resolution in the FB-OCC (Li et al., 2023c) open-source code to ensure a fair evaluation. Compared to previous methods, our approach demonstrates superior performance in terms of mIoU, particularly excelling in medium and small foreground classes (such as barrier, pedestrian, bicycle, and car). This highlights that using a voting-based enhancement mechanism for handling foreground classes aligns better with scene characteristics and enhances the ability for semantic filling within objects. Additionally, our method performs exceptionally well in medium and large scene structure classes (such as driveable surface and sidewalk), emphasizing that utilizing a cross state space module for the fusion and interaction of interframe 3D voxel features is more in line with the temporal variations of the scene, thereby enhancing the ability to leverage temporal information.

**LiDAR Semantic Segmentation.** To better assess the effectiveness of our method, we project the 3D semantic occupancy prediction results onto the point cloud and conduct comparative experiments on the LiDAR semantic segmentation task. As shown in Table 2, we compare our results with those of other 3D semantic occupancy prediction meth-

*Table 1.* **3D semantic occupancy prediction performance on Occ3D-nuScenes dataset.** In all methods, only a single-modal camera image is used as input during testing, without any additional augmentations or model ensembles incorporated. "4f" and "8f" refer to the model's input containing data from 4 frames or 8 frames, respectively, incorporating temporal information. The symbol "*" marks that these models use the visible mask during training.

| METHODS | IMAGE BACKBONE | PUBLICATION | mIoU↑ | OTHERS↑ | BARRIER↑ | BICYCLE↑ | BUS↑ | CAR↑ | CONST. VEH.↑ | MOTORCYCLE↑ | PEDESTRIAN↑ | TRAFFIC CONE↑ | TRAILER↑ | TRUCK↑ | DRIVE. SUF.↑ | OTHER FLAT↑ | SIDEWALK↑ | TERRAIN↑ | MANMADE↑ | VEGETATION↑ |
|---|---|---|---|---|---|---|---|---|---|---|---|---|---|---|---|---|---|---|---|---|
| MONOSCENE (CAO & DE CHARETTE, 2022) | R101 | CVPR'22 | 6.06 | 1.75 | 7.23 | 4.26 | 4.93 | 9.38 | 5.67 | 3.98 | 3.01 | 5.90 | 4.45 | 7.17 | 14.91 | 6.32 | 7.92 | 7.43 | 1.01 | 7.65 |
| BEVDET (HUANG ET AL., 2021) | R101 | ARXIV'21 | 11.73 | 2.09 | 15.29 | 0.00 | 4.18 | 12.97 | 1.35 | 0.00 | 0.43 | 0.13 | 6.59 | 6.66 | 52.72 | 19.04 | 26.45 | 21.78 | 14.51 | 15.26 |
| OCCFORMER (ZHANG ET AL., 2023B) | R101 | ICCV'23 | 21.93 | 5.94 | 30.29 | 12.32 | 34.40 | 39.17 | 14.44 | 16.45 | 17.22 | 9.27 | 13.90 | 26.36 | 50.99 | 30.96 | 34.66 | 22.73 | 6.76 | 6.97 |
| TPVFORMER (HUANG ET AL., 2023) | R101 | CVPR'23 | 28.34 | 6.67 | 39.20 | 14.24 | 41.54 | 46.98 | 19.21 | 22.64 | 17.87 | 14.54 | 30.20 | 35.51 | 56.18 | 33.65 | 35.69 | 31.61 | 19.97 | 16.12 |
| CTF-OCC (TIAN ET AL., 2024) | R101 | NIPS'23 | 28.53 | 8.09 | 39.33 | 20.56 | 38.29 | 42.24 | 16.93 | 24.52 | 22.72 | 21.05 | 22.98 | 31.11 | 53.33 | 33.84 | 37.98 | 33.23 | 20.79 | 18.00 |
| BEVFORMER (4F) (LI ET AL., 2022)* | R101 | ECCV'22 | 39.24 | 10.13 | 47.91 | 24.9 | 47.57 | 54.52 | 20.23 | 28.85 | 28.02 | 25.73 | 33.03 | 38.56 | 81.98 | 40.65 | 50.93 | 53.02 | 43.86 | 37.15 |
| BEVDET (8F) (HUANG & HUANG, 2022)* | SWIN-B | ARXIV'22 | 42.02 | 12.15 | 49.63 | 25.10 | 52.02 | 54.46 | 27.87 | 27.99 | 28.94 | 27.23 | 36.43 | 42.22 | 82.31 | 43.29 | 54.62 | 57.90 | 48.61 | 43.55 |
| PANOOCC (4F) (WANG ET AL., 2024)* | R101 | CVPR'24 | 42.13 | 11.67 | 50.48 | 29.64 | 49.44 | 55.52 | 23.29 | 33.26 | 30.55 | 30.99 | 34.43 | 42.57 | 83.31 | 44.23 | 54.40 | 56.04 | 45.94 | 40.40 |
| FB-OCC (8F) (LI ET AL., 2023C)* | R50 | ICCV'23 | 40.67 | 10.48 | 47.54 | 29.62 | 45.76 | 50.15 | 28.45 | 28.56 | 28.09 | 29.51 | 34.32 | 37.44 | 80.69 | 46.51 | 54.36 | 57.14 | 44.95 | 39.02 |
| FB-OCC (8F) (LI ET AL., 2023C)* | R101 | ICCV'23 | 42.24 | 10.98 | 49.06 | 31.09 | 47.33 | 51.72 | 30.02 | 30.08 | 29.66 | 31.16 | 35.89 | 39.05 | 82.18 | 48.08 | 55.96 | 58.71 | 46.52 | 40.59 |
| OCTREEOCC (4F) (LU ET AL., 2023)* | R101 | NIPS'24 | 44.02 | 11.96 | 51.70 | 29.93 | 53.52 | 56.77 | 30.83 | 33.17 | 30.65 | 29.99 | 37.76 | 43.87 | 83.17 | 44.52 | 55.45 | 58.86 | 49.52 | 46.33 |
| CSV-OCC-S (4F)* | R50 | OURS | 42.37 | 11.75 | 50.65 | 27.29 | 50.98 | 55.12 | 29.94 | 30.98 | 29.70 | 30.58 | 35.69 | 42.64 | 82.51 | 44.58 | 54.96 | 56.92 | 45.97 | 40.08 |
| CSV-OCC-B (4F)* | R101 | OURS | 44.93 | 12.98 | 53.12 | 32.76 | 54.13 | 58.24 | 31.01 | 35.07 | 33.14 | 33.49 | 37.05 | 44.17 | 84.02 | 47.11 | 56.17 | 59.16 | 48.69 | 43.57 |

*Table 2.* **LiDAR semantic segmentation results on nuScenes validation set.** "4f" refer to the model's input containing data from 4 frames, incorporating temporal information. In Modality, "L" and "C" represent LiDAR input and camera input respectively.

| METHODS | IMAGE BACKBONE | MODALITY | PUBLICATION | mIoU↑ | BARRIER↑ | BICYCLE↑ | BUS↑ | CAR↑ | CONST. VEH.↑ | MOTORCYCLE↑ | PEDESTRIAN↑ | TRAFFIC CONE↑ | TRAILER↑ | TRUCK↑ | DRIVE. SUF.↑ | OTHER FLAT↑ | SIDEWALK↑ | TERRAIN↑ | MANMADE↑ | VEGETATION↑ |
|---|---|---|---|---|---|---|---|---|---|---|---|---|---|---|---|---|---|---|---|---|
| RANGENET++ (MILIOTO ET AL., 2019) | - | L | IROS'19 | 65.5 | 66.0 | 21.3 | 77.2 | 80.9 | 30.2 | 66.8 | 69.6 | 52.1 | 54.2 | 72.3 | 94.1 | 66.6 | 63.5 | 70.1 | 83.1 | 79.8 |
| POLARNET (ZHANG ET AL., 2020) | - | L | CVPR'20 | 71.0 | 74.7 | 28.2 | 85.3 | 90.9 | 35.1 | 77.5 | 71.3 | 58.8 | 57.4 | 76.1 | 96.5 | 71.1 | 74.7 | 74.0 | 87.3 | 85.7 |
| SALSANEXT (CORTINHAL ET AL., 2020) | - | L | ISVC'20 | 72.2 | 74.8 | 34.1 | 85.9 | 88.4 | 42.2 | 72.4 | 72.2 | 63.1 | 61.3 | 76.5 | 96.0 | 70.8 | 71.2 | 71.5 | 86.7 | 84.4 |
| CYLINDER3D++ (ZHU ET AL., 2021) | - | L | CVPR'21 | 76.1 | 76.4 | 40.3 | 91.2 | 93.8 | 51.3 | 78.0 | 78.9 | 64.9 | 62.1 | 84.4 | 96.8 | 71.6 | 76.4 | 75.4 | 90.5 | 87.4 |
| RPVNET (XU ET AL., 2021) | - | L | ICCV'21 | 77.6 | 78.2 | 43.4 | 92.7 | 93.2 | 49.0 | 85.7 | 80.5 | 66.0 | 66.9 | 84.0 | 96.9 | 73.5 | 75.9 | 76.0 | 90.6 | 88.9 |
| BEVFORMER (LI ET AL., 2022) | R101 | C | ECCV'22 | 56.2 | 54.0 | 22.8 | 76.7 | 74.0 | 45.8 | 53.1 | 44.5 | 24.7 | 54.7 | 65.5 | 88.5 | 58.1 | 50.5 | 52.8 | 71.0 | 63.0 |
| TPVFORMER (HUANG ET AL., 2023) | R101 | C | CVPR'23 | 68.9 | 70.0 | 40.9 | 93.7 | 85.6 | 49.8 | 68.4 | 59.7 | 38.2 | 65.3 | 83.0 | 93.3 | 64.4 | 64.3 | 64.5 | 81.6 | 79.3 |
| PANOOCC (4F) (WANG ET AL., 2024) | R50 | C | CVPR'24 | 68.1 | 70.7 | 37.9 | 92.3 | 85.0 | 50.7 | 64.3 | 59.4 | 35.3 | 63.8 | 81.6 | 94.2 | 66.4 | 64.8 | 68.0 | 79.1 | 75.6 |
| PANOOCC (4F) (WANG ET AL., 2024) | R101 | C | CVPR'24 | 71.6 | 74.3 | 43.7 | 95.4 | 87.0 | 56.1 | 64.6 | 66.2 | 41.4 | 71.5 | 85.9 | 95.1 | 70.1 | 67.0 | 68.1 | 80.9 | 77.4 |
| CSV-OCC-S (4F) | R50 | C | OURS | 70.3 | 71.9 | 41.7 | 92.1 | 86.8 | 51.2 | 62.5 | 63.0 | 41.1 | 69.1 | 85.7 | 95.5 | 67.4 | 69.4 | 66.5 | 85.8 | 75.3 |
| CSV-OCC-B (4F) | R101 | C | OURS | 73.4 | 74.4 | 44.1 | 95.6 | 89.5 | 53.9 | 65.8 | 67.5 | 45.2 | 72.6 | 88.5 | 97.2 | 71.9 | 72.4 | 69.8 | 87.3 | 78.9 |

ods and LiDAR semantic segmentation methods based on point cloud inputs on the nuScenes validation set. Our method achieves the best mIoU performance among the image-based 3D semantic occupancy prediction methods, demonstrating excellent results for specific semantic classes (such as pedestrian, bicycle, and car). This success can be attributed to the feature correction and interaction capabilities of the voting-based enhancement mechanism and the cross-state space module. In comparison with the LiDAR semantic segmentation methods based on point cloud inputs, we have approached the performance of current state-of-the-art lidar-based methods using purely image input.

### 4.3. Ablation Study

We perform ablation experiments to evaluate the design choices of CSV-Occ on Occ3D-nuScenes (Tian et al., 2024) validation set. By default, we utilize the CSV-Occ-B configuration.

**Different Multi-frame Feature Aggregation Methods.** Table 3 compares the performance of different multi-frame voxel feature aggregation methods used in the TVT module. #b is the default setting. By observing #A, #B, #C, and #D, we find that adjusting the state dimension in the Cross SSM affects the mIoU, with the best performance achieved when the state dimension is set to 4. By comparing #B, #E, #F, and #G, it is discovered that Cross SSM outperforms the utilization of temporal self-attention (Li et al., 2022), MLP mixer (Tolstikhin et al., 2021), or direct concatenation. This indicates that Cross SSM is more capable of concentrating on the differences and similarities between frames, making it more suitable for the integration and interaction of inter-frame voxel features.

**Time-based View Transformation and Voting-based Enhancement Mechanism.** Table 4 verifies that TVT and VEM can have positive effects on the model. Compared with the single-frame model, the multi-frame training and inference model using TVT performs better in the 3D semantic occupancy prediction and LiDAR semantic segmentation tasks. VEM further enhances the interaction between object surface and internal feature learning, and also improves the performance of both tasks. The effect of VEM is shown in Fig. 7, and it can eliminate or greatly alleviate IOV.

**The Weights of the Loss Function.** Table 5 shows the comparison of various combinations of different loss function weights. Comparing $\lambda_{RCR}$ in #A and #C, #B and #D, it can be found that $L_{RCR}$ can bring significant gains to the

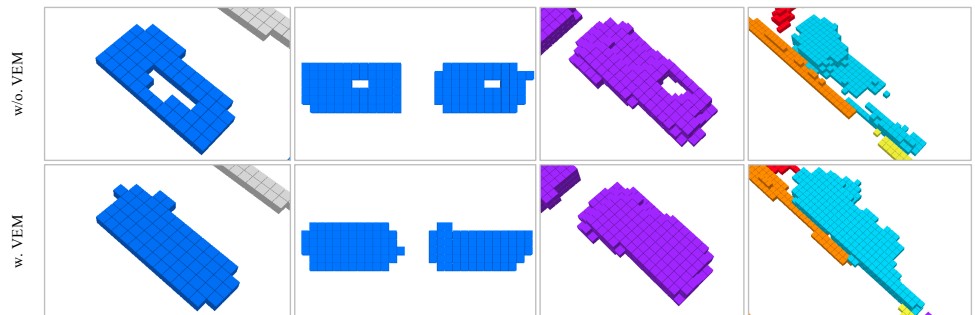

*Figure 7.* **Visualizations of the occupancy predictions IOV with and without VEM.**

*Table 3.* **Ablation study for different multi-frame voxel feature aggregation methods.** TSA stands for the temporal self-attention structure.

| # | FUSION MODULE | STATE DIMENSION | MIOU (OCC.)$^\uparrow$ | MIOU (SEG.)$^\uparrow$ | TRAINABLE PARAMS (M)$^\downarrow$ |
|---|---|---|---|---|---|
| A | | 2 | 44.77 | 73.05 | **68.3** |
| B | CROSS SSM (OURS) | 4 | **44.93** | **73.42** | 68.5 |
| C | | 8 | 44.89 | 73.41 | 68.9 |
| D | | 16 | 43.58 | 72.70 | 69.6 |
| E | TSA (LI ET AL., 2022) | - | 43.17 | 72.62 | 69.3 |
| F | MLP-MIXER (TOLSTIKHIN ET AL., 2021) | - | 43.22 | 72.67 | 77.5 |
| G | CONCAT. AND CONV. | - | 42.31 | 72.53 | 68.8 |

*Table 4.* **Effects of time-based view transformation and voting-based enhancement mechanism.** When TVT is not used, we replace it with SCA and TSA of BEVFormer (Li et al., 2022).

| # | TVT | VEM | MIOU (OCC.)$^\uparrow$ | MIOU (SEG.)$^\uparrow$ |
|---|---|---|---|---|
| A | - | - | 40.37 | 69.81 |
| B | ✓ | - | 43.71 | 72.77 |
| C | - | ✓ | 42.08 | 71.33 |
| D | ✓ | ✓ | **44.93** | **73.42** |

*Table 5.* **Ablation for the weights of the loss function.**

| # | $\lambda_{RCR}$ | $\lambda_{focal}$ | $\lambda_{lov}$ | $\lambda_{aff}$ | MIOU (OCC.)$^\uparrow$ | MIOU (SEG.)$^\uparrow$ |
|---|---|---|---|---|---|---|
| A | | 2.0 | - | - | 40.61 | 69.79 |
| B | - | 2.0 | 2.0 | 1.5 | 43.98 | 72.80 |
| C | | 2.0 | - | - | 42.83 | 71.91 |
| D | 1.0 | 2.0 | 2.0 | 1.5 | **44.93** | **73.42** |
| E | | 5.5 | 2.0 | 1.5 | 44.78 | 73.30 |
| F | | 2.0 | 2.0 | 4.0 | 44.04 | 72.95 |

performance, which also means that explicitly supervising the voxel offsets can enable the model to more clearly learn the physical meaning of the center voting. We also conducted experiments on various combinations of different loss function weights and found that the performance is the best when $\lambda_{RCR} = 1$, $\lambda_{focal} = 2$, $\lambda_{lov} = 2$, and $\lambda_{aff} = 1.5$.

## 4.4. Temporal Computation Complexity

The following efficiency experiments were all measured with the batch size set to 1.

Fig. 8 shows the impact of changes in BEV size on inference speed and memory. We explored how initialized voxel query size impacts model efficiency. Keeping its height at 4, we adjusted the BEV side length from 25 to 200 (max $200 \times 200$, not exceeding occupancy ground truth). The left figure shows samples inferred per second; the right shows inference memory consumption. As voxel query size grows, especially past 100, TSA and MLP mixer's inference speed and memory consumption worsen. Cross SSM's efficiency decline is more stable due to its linear computational complexity. A larger query size means a longer flattened token sequence, and Cross SSM needs only one scan for multi frame fusion.

Fig. 9 shows how the number of inference frames affects model efficiency. The initialized voxel query size is set at $100 \times 100 \times 4$. Cross SSM outperforms in both inference

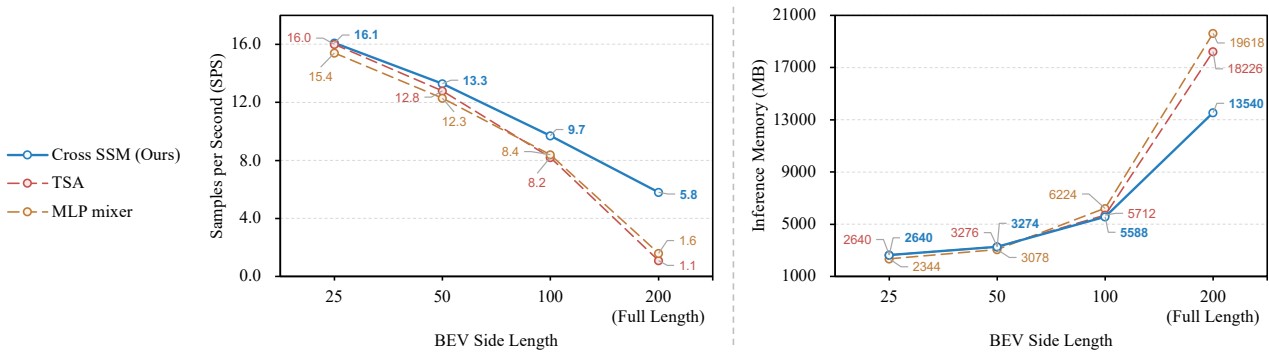

*Figure 8.* **The impact of changes in BEV size on inference speed and memory.**

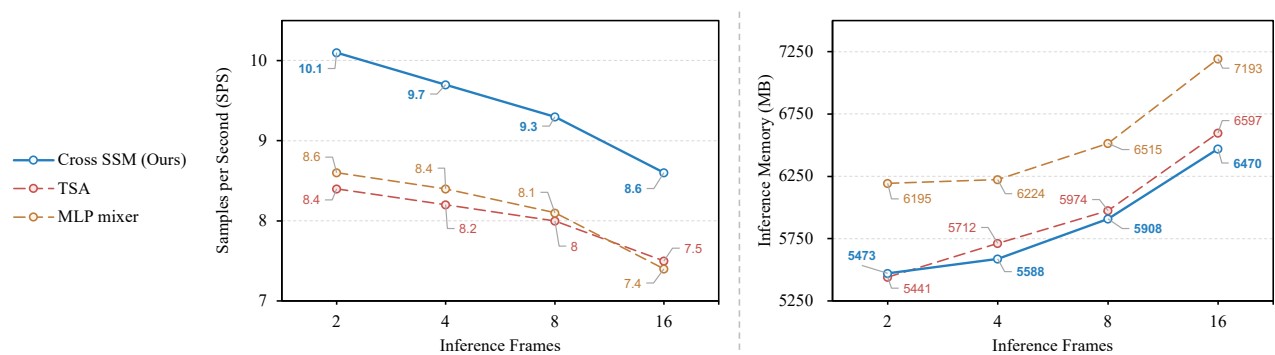

*Figure 9.* **The impact of changes in the number of inference frames on inference speed and memory.**

speed and memory consumption. Still, as the number of inference frames rises, Cross SSM's efficiency change trend is like that of TSA and MLP mixer. In CSV-Occ, the number of inference frames equals the number of multi-frame fusion module calls, and the feature sequence length per call depends only on the voxel query size. So, more inference frames don't give Cross SSM a trend advantage.

## 5. Conclusion

In this paper, we propose CSV-Occ, a novel camera-based 3D semantic occupancy prediction method. CSV-Occ integrates temporal information through an extended state space model and enhances feature representation in a coarse-to-fine manner by explicitly predicting the instance to which voxels belong, which achieves a comprehensive understanding of the scene. Extensive experimental results on the nuScenes dataset and Occ3D-nuScenes confirm the effectiveness of CSV-Occ and its potential in advancing 3D semantic occupancy prediction. In our view, 3D semantic occupancy representation is a promising new paradigm for

future 3D scene perception.

## Impact Statement

This paper presents work whose goal is to advance the field of Machine Learning. There are many potential societal consequences of our work, none which we feel must be specifically highlighted here.

## Acknowledgements

This work was supported in part by National Natural Science Foundation of China under Grant 62476088, in part by the Shanghai Automotive Industry Science and Technology Development Foundation under Grant 2304, and in part by the Science and Technology Commission of Shanghai Municipality under Grant 20DZ2254400.

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

## A. Details of Time-based View Transformation

Specifically, as shown in Fig. 3, we initialize a learnable 3D voxel query tensor $Q \in \mathbb{R}^{h \times w \times l \times d_{\text{voxel}}}$ as the basis for 3D spatial information aggregation. According to the given intrinsic and extrinsic parameters of the $N$ cameras, we attempt to project the center coordinate of each voxel in the 3D voxel query onto the two-dimensional coordinate system of the $N$ camera images (if there is view overlap in the surround-camera images, then some 3D voxels may be projected onto multiple views simultaneously) to obtain voxel-pixel pairs. Then, we sample the features around these projected 2D coordinates and achieve the aggregation of 2D features to 3D voxels through the deformable attention (Zhu et al., 2020) mechanism:

$$F^p = \frac{1}{|\mathcal{V}_{\text{hit}}|} \sum_{i \in \mathcal{V}_{\text{hit}}} \text{DefAttn}\left(Q^p, \mathcal{P}\left(q^p, i\right), X_i\right) \tag{10}$$

Where $F^p$ and $Q^p$ denote the $p$-th element of the output feature and the 3D volume query, respectively. $q^p$ is the corresponding 3D coordinate of $Q^p$, $\mathcal{P}$ is the 3D-to-2D projection function including intrinsic and extrinsic parameters, $X_i$ is the feature map of the $i$-th camera. $\mathcal{V}_{\text{hit}}$ represents the set of hit view indices projected by $q^p$. $\text{DefAttn}$ represents deformable attention computation (Zhu et al., 2020).

## B. Derivations of Cross State Space Module

The length and feature dimension of the input sequence are represented by $S$ and $d$ respectively. The original Attention formula is as follows:

$$\text{Attention}(\bar{Q}, \bar{K}, V) = \text{softmax}(\frac{\bar{Q}\bar{K}^\top}{\sqrt{d}})V \tag{11}$$

Where $\bar{Q} \in \mathbb{R}^{S \times d}$, $\bar{K} \in \mathbb{R}^{S \times d}$, and $V = \{v_1, \ldots, v_S\} \in \mathbb{R}^{S \times d}$ are obtained by mapping the same input feature sequence tensor through three independent MLP mappings (in Cross Attention, $\bar{Q}$ can be obtained by mapping another input feature sequence tensor). For the convenience of subsequent derivation, we abandon the tensor value scaling and softmax, and we can obtain a new calculation:

$$\bar{Y} = \bar{Q}\bar{K}^\top V \tag{12}$$

Where $\bar{Y} \in \mathbb{R}^{S \times d}$ represents the output of the new calculation. Particularly, the $j$-th slice of $\bar{Y}$ along the feature channel dimension $d$, denoted as $\bar{Y}^{[j]} \in \mathbb{R}^{S \times 1}$, can be written as:

$$\bar{Y}^{[j]} = \bar{Q}\bar{K}^\top V^{[j]} \tag{13}$$

We introduce the intermediate variable tensors $\mathbf{W} = \{w_1, \ldots, w_S\} \in \mathbb{R}^{S \times d \times d}$ and $E \in \mathbb{R}^{S \times S}$, where the role of $E$ is similar to the Attention mask, and $E$ is a full 1 matrix to ensure the equality still holds. Meanwhile, let $Q = \frac{\bar{Q}}{W^{[j]}} = \{q_1, \ldots, q_S\}$ and $K = \bar{K} \odot W^{[j]} = \{k_1, \ldots, k_S\}$, and we can obtain:

$$\bar{Y}^{[j]} = \left[\left(Q \odot \mathbf{W}^{[j]}\right)\left(\frac{K}{\mathbf{W}^{[j]}}\right)^\top \odot E\right] V^{[j]} \tag{14}$$

Where $\odot$ represents the element-wise multiplication of tensors, and the division sign also represents the element-wise division of tensors. Introduce the variable tensor $\mathbf{H} = \{h_1, \ldots, h_S\} \in \mathbb{R}^{S \times d \times d}$, construct a new term and add it to the original term. To ensure the equality holds, multiply the new term by the all-zero tensor $O \in \mathbb{R}^{S \times d}$:

$$\bar{Y}^{[j]} = \left[\left(Q \odot \mathbf{W}^{[j]}\right) \odot O\right] h_1^{[j]} + \left[\left(Q \odot \mathbf{W}^{[j]}\right)\left(\frac{K}{\mathbf{W}^{[j]}}\right)^\top \odot E\right] V^{[j]} \tag{15}$$

Maintaining the same equality structure, replace $O$ and $E$ with $E$ and $M$ respectively while keeping the shape unchanged, where the element values of the lower triangular part of $M$ are 1 and the rest are 0, to obtain $Y^{[j]} \in \mathbb{R}^{S \times 1}$, where $Y = \{y_1, \ldots, y_S\}$:

$$Y^{[j]} = \left[ \left( Q \odot \mathbf{W}^{[j]} \right) \odot E \right] h_1^{[j]} + \left[ \left( Q \odot \mathbf{W}^{[j]} \right) \left( \frac{K}{\mathbf{W}^{[j]}} \right)^{\top} \odot M \right] V^{[j]} \tag{16}$$

The $t$-th slice of $Y^{[j]}$ along the feature sequence dimension $S$, denoted as $y_t^{[j]} \in \mathbb{R}^{1 \times 1}$:

$$y_t^{[j]} = \left( q_t \odot w_t^{[j]} \right) h_1^{[j]} + \sum_{i=1}^{t} \left( \frac{q_t \odot w_t^{[j]}}{w_i^{[j]}} k_i^{\top} \right) \odot v_i^{[j]} \tag{17}$$

Similarly, the $t$-th slice of $Y$ along the feature sequence dimension $S$, denoted as $y_t \in \mathbb{R}^{1 \times d}$, and extract $q_t$:

$$\begin{aligned}
y_t &= q_t \left( w_t \odot h_1 \right) + q_t \sum_{i=1}^{t} \frac{w_t}{w_i} \odot \left( k_i^{\top} v_i \right) \\
&= q_t \left[ w_t \odot h_1 + w_t \odot \sum_{i=1}^{t} \frac{k_i^{\top} v_i}{w_i} \right] \\
&= q_t h_t \\
h_t &= w_t \odot h_1 + w_t \odot \sum_{i=1}^{t} \frac{k_i^{\top} v_i}{w_i}
\end{aligned} \tag{18}$$

We define $w_t = \prod_{l=1}^{t} e^{\mathbf{A}\Delta_l}$, and substitute it for calculation, then transform $h_t$ into the recursive form:

$$\begin{aligned}
h_t &= \prod_{l=1}^{t} e^{A\Delta_l} \odot h_1 + \prod_{l=1}^{t} e^{A\Delta_l} \odot \sum_{i=1}^{t} \frac{k_i^{\top} v_i}{\prod_{l=1}^{i} e^{A\Delta_l}} \\
&= e^{A(\Delta_1 + \ldots + \Delta_t)} \odot h_1 + e^{A(\Delta_1 + \ldots + \Delta_t)} \odot \sum_{i=1}^{t} \frac{k_i^{\top} v_i}{e^{A(\Delta_1 + \ldots + \Delta_i)}} \\
&= e^{A\Delta_t} \odot h_{t-1} + k_t^{\top} v_t
\end{aligned} \tag{19}$$

Substitute $V = X \odot \Delta$, $K = B$, $Q = C$, where $X = \{x_1, \ldots, x_S\} \in R^{S \times d}$ is the input feature sequence tensor, $\Delta \in R^{S \times d}$, and add a skip connection to $y_t$ to obtain the following formula in the form of a state space model:

$$\begin{cases} h_t = e^{A\Delta_t} \odot h_{t-1} + B_t^{\top} \left( x_t \odot \Delta_t \right) \\ y_t = C_t h_t + D \odot x_t \end{cases} \tag{20}$$

To sum up, we believe that the state space model can be regarded as a special subset of the Attention model, which makes it possible for the state space model to achieve feature interaction of two different input sequence tensors. And observing the above derivation, it is not difficult to find that $\bar{Q}$ (in Cross Attention, $\bar{Q}$ can be obtained by mapping another input feature sequence tensor) is gradually replaced by $C$ in the process, which means that we believe that the $C$ variable in the state space model can also be similarly obtained by mapping another input feature sequence tensor. Thus, we constructed Cross SSM (Sec. 3.3.1 of the main text).

## C. Datasets and Evaluation Metrics

**The nuScenes Dataset** (Caesar et al., 2020) comprises 1000 scene sequences, distributed as 700 in the training set, 150 in the validation set, and 150 in the test set. Each scene sequence is recorded at a 20Hz frequency and lasts for 20 seconds.

Each sample contains RGB images from 6 surround cameras and point cloud data from a LiDAR sensor with 32 beams. For object detection task labels, key samples are annotated at a rate of 2Hz with ground truth labels for 10 foreground object classes, including 3D bounding boxes and categories. In addition to the 10 foreground classes, semantic segmentation task labels also encompass 6 background classes represented in per-point labels within the point cloud.

**The Occ3D-nuScenes Dataset** (Tian et al., 2024) is an extension of the nuScenes dataset that expands the semantic occupancy prediction task label set. In the egocentric coordinate system, the semantic occupancy space ranges from -40m to 40m on the X and Y axes, and from -1m to 5.4m on the Z axis. The voxel size of the occupancy label is $0.4m \times 0.4m \times 0.4m$ (resulting in a spatial resolution of $200 \times 200 \times 16$ for occupancy labels). The semantic labels include 17 categories, comprising 16 categories specifically designated for semantic segmentation tasks, as well as an "other" category. Additionally, visible masks are supplied for both the LiDAR and camera modalities.

**Evaluation Metrics.** We use the mean Intersection over Union (mIoU) as the evaluation metric for assessing both LiDAR semantic segmentation and 3D semantic occupancy prediction tasks:

$$mIoU = \frac{1}{C} \sum_{i=1}^{C} \frac{TP_i}{TP_i + FP_i + FN_i}, \tag{21}$$

In this equation, $TP_i$, $FP_i$, and $FN_i$ represent the counts of true positives, false positives, and false negatives for class $i$, respectively, with $C$ denoting the total number of semantic classes.

## D. Experimental results

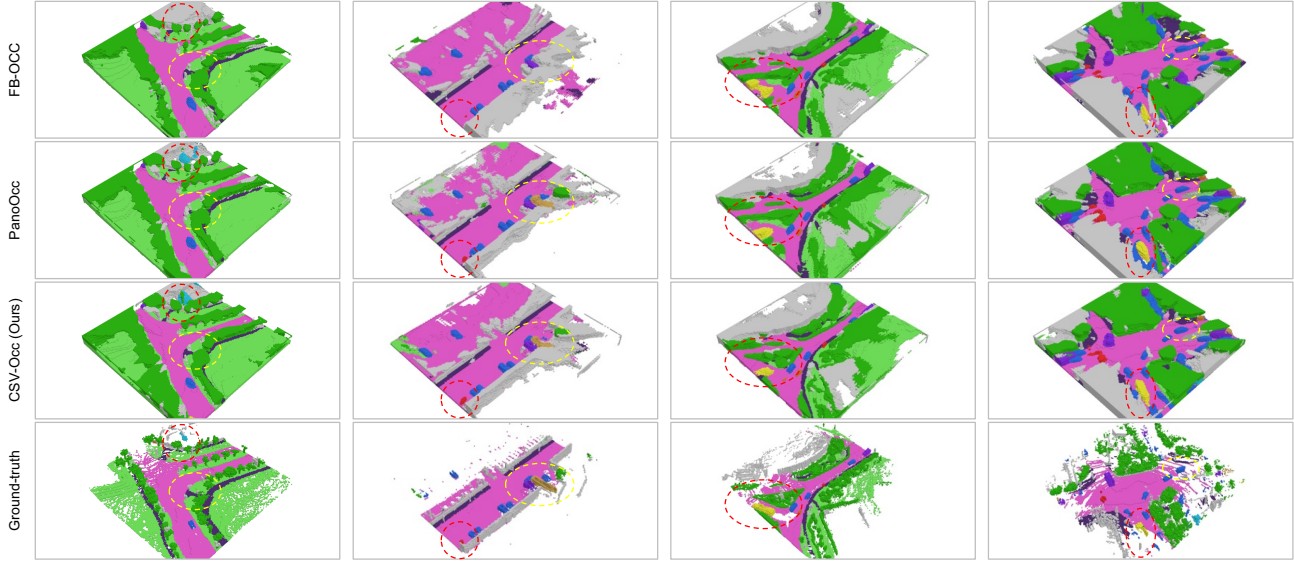

*Figure 10.* **Qualitative comparison of 3D semantic occupancy prediction.**

**Visualization.** Fig. 10 shows the qualitative results of our method and some other state-of-the-art methods on the Occ3D-nuScenes dataset, demonstrating that our method can understand the overall structure of the scene more completely and capture finer-grained geometric details.

**Design of Time-based View Transformation.** In the TVT module, we initialize a learnable voxel query to convert image features into voxel space. Table 6 compares the impact of different voxel query sizes on the experimental results. #H is the default setting for our method. In settings #C, #D, #E, and #F, we changed the height dimension while fixing the BEV size. It was observed that the coding density of height information significantly affects the performance of semantic occupancy prediction and segmentation tasks. Additionally, under a fixed number of voxels, we designed experiments where (1) #A and #C have 5k voxels, (2) #B and #D have 10k voxels, (3) #E and #G have 20k voxels, and (4) #F and #H have 40k voxels.

*Table 6.* **Ablation study for different voxel query sizes.** The improvement of BEV information density is very conducive to improving the fineness of realizing 3D scene understanding.

| # | VOXEL QUERY SIZE | | VOXEL NUMBER | MIOU (OCC.)↑ | MIOU (SEG.)↑ |
|---|---|---|---|---|---|
| | BEV SIZE | HEIGHT | | | |
| A | 25 × 25 | 8 | 5K | 40.88 | 68.97 |
| B | | 16 | 10K | 42.38 | 71.91 |
| C | 50 × 50 | 2 | 5K | 40.37 | 68.83 |
| D | | 4 | 10K | 42.51 | 71.93 |
| E | | 8 | 20K | 44.08 | 72.35 |
| F | | 16 | 40K | 44.91 | 73.06 |
| G | 100 × 100 | 2 | 20K | 44.16 | 72.63 |
| H | | 4 | 40K | **44.93** | **73.42** |

*Table 7.* **Ablation study for varying configurations for the number of aggregated frames and frame intervals.** Key sample images are sampled at 2Hz (with a frame interval of 0.5s) in the Occ3D-nuScenes dataset. The time span can be calculated by the number of fused frames and the number of frame intervals.

| # | FRAMES | FRAME INTERVALS | TIME SPAN | MIOU (OCC.)↑ | MIOU (SEG.)↑ |
|---|---|---|---|---|---|
| A | 4 | 1 | 1.5s | 44.18 | 72.98 |
| B | | 2 | 3.0s | **44.93** | **73.42** |
| C | | 4 | 6.0s | 44.61 | 73.11 |
| D | 2 | 1 | 0.5s | 43.59 | 72.42 |
| E | | 2 | 1.0s | 43.74 | 72.49 |
| F | | 4 | 2.0s | 43.97 | 72.56 |
| G | | 6 | 3.0s | 44.02 | 72.68 |
| H | 1 | - | - | 42.08 | 71.33 |

The results indicate that only the former in experiment (1) outperforms the latter, suggesting that the improvement in mIoU from enhancing the encoding density of height information is much smaller than that from increasing the encoding density of BEV information. This finding highlights the importance of fine-grained encoding of BEV information for achieving comprehensive scene understanding.

**The Number of Frames and Frame Intervals.** As mentioned above, one of the functions of the TVT module is to aggregate multi-frame voxel features. Table 7 shows the impact of different settings for the number of aggregated frames and frame intervals on model performance in the TVT module. #B is the default setting of our method. By comparing #A, #B, and #C, we find that #B (which selects a frame interval of length 2 while keeping the number of aggregated frames at 4) can enhance the performance of the TVT module. However, observations from #D, #E, #F, and #G yield different results, with #G (which selects a frame interval of length 6 while keeping the number of aggregated frames at 2) exhibiting the highest performance. From this, we observe that the time spans of #B and #G are closer, indicating that the time span's impact on model performance is more significant. Comparing time spans that are relatively close, (1) #B, #F, and #G have time spans between 2s and 3s, while (2) #A and #E have time spans between 1s and 1.5s. We find that under similar time spans, an increase in the number of aggregated frames often leads to a better mIoU, which underscores the importance of temporal information density. At the same time, we also discover that extending the time span does not infinitely improve model performance; in fact, when the time span is too long, the mIoU of dynamic objects tends to decrease. This is due to the dynamic ambiguity that arises from the information and features of dynamic objects over extended time spans.

**Design of Parallel Interaction Strategy.** In Table 8, we conducted an ablation study on the effectiveness of the parallel interaction strategy (PIS). #A is a simple per-voxel baseline that processes features directly through a series of MLPs. #B introduces the parallel interaction strategy without feature descriptor sorting, which improves performance by 1.5 occupancy mIoU and 0.56 segmentation mIoU. To enhance the fusion efficiency of highly correlated features, we implemented a feature descriptor sorting mechanism in #C and #D, which further increased inference accuracy, ultimately achieving the highest

*Table 8.* **Ablation study for the parallel interaction strategy.** The symbol ✓ represents the use of parallel interaction strategy.

| # | PIS | SORTING | MIOU (OCC.)↑ | MIOU (SEG.)↑ |
|---|-----|---------|-------------|-------------|
| A | - | - | 42.18 | 72.15 |
| B | ✓ | - | 43.68 | 72.71 |
| C | ✓ | FORWARD | **44.93** | **73.42** |
| D | ✓ | REVERSE | 44.86 | 73.37 |

44.93 occupancy mIoU and 73.42 segmentation mIoU with forward sorting (compared to 44.86 occupancy mIoU and 73.37 segmentation mIoU with reverse sorting).

## E. Limitation

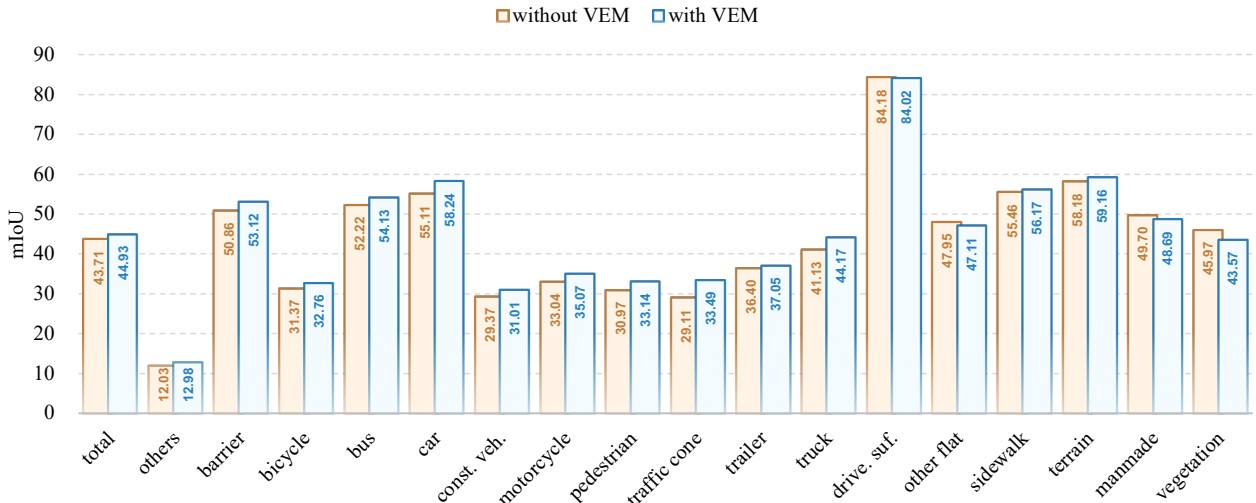

*Figure 11.* **Performance comparison of each category with and without VEM.**

**Lower Performance on Certain Background Classes.** Through statistics, we've found that the low performance of our method in certain background categories is primarily attributed to the VEM module. As depicted in Fig. 11, upon enabling the voting mechanism, the performance of all foreground categories has increased. Conversely, a decline in performance is observed in four background categories: driveable surface, other flat, manmade, and vegetation.

This occurs because VEM relies on the coarse semantic occupancy prediction to divide foreground category voxels. When the coarse semantic occupancy prediction is inaccurate, VEM may predict relative centers for some background voxels as well. The occupancy of background categories usually has a large volume and lacks distinct instance centers. This leads to training confusion in VEM. During inference, it wrongly updates the features of background voxels, causing the occupancy classification head to give incorrect category predictions. As a result, the number of background voxels decreases, which directly impacts the prediction performance of background categories.

We also calculated the proportion of the number of foreground and background voxels predicted by the model with and without VEM. As shown in Fig. 12, it can be seen that VEM can significantly increase the number of foreground voxels. This partly indicates that VEM can improve the Internal Occupancy Vacancy (IOV) situation and successfully predict free voxels as foreground voxels. However, we unexpectedly found that the number of background voxels decreased, which we think is related to the training confusion caused by VEM.

**Weak in Instance-Level Occupancy Prediction.** CSV-Occ can generate instance-level results, but requires additional

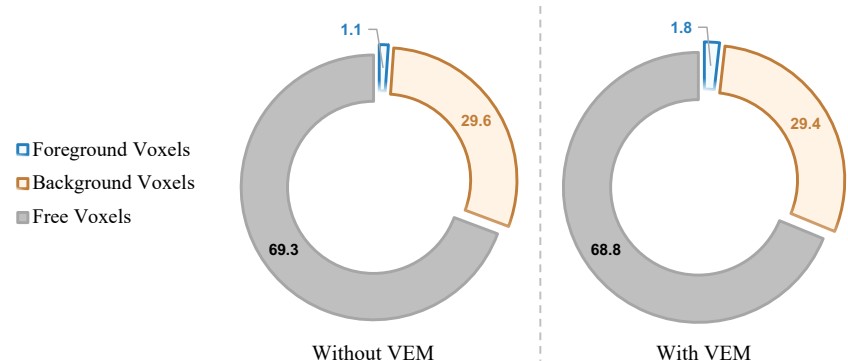

*Figure 12.* **The proportion of the number of foreground and background voxels predicted by the model with and without VEM.**

*Table 9.* **Comparison of instance-level occupancy prediction.**

| METHOD | MODALITY | PUBLICATION | PQ | SQ | RQ |
|---|---|---|---|---|---|
| PANOPTICTRACKNET (HURTADO ET AL., 2020) | L | ARXIV'20 | 51.4 | 80.2 | 63.3 |
| EFFICIENTLPS (SIROHI ET AL., 2021) | L | I-TR'21 | 62.0 | 83.4 | 73.9 |
| LIDARMULITINET (YE ET AL., 2023) | L | AAAI'23 | **81.8** | **89.7** | **90.8** |
| PANOOCC (WANG ET AL., 2024) | C | CVPR'24 | **62.1** | **82.1** | **75.1** |
| CSV-OCC-INSTANCE | C | OURS | 48.3 | 79.6 | 60.5 |

post-processing. Our implementation involves three key steps: (1) Applying Relative Central Regression to the final semantic occupancy output for voxel-level center prediction (2) Clustering potentially scattered center predictions (Due to the error in the center voting predicted by the model, multiple discrete center points often appear) to form coherent instances (3) The clustered central voxels then determine instance ID assignments through voting relationships. Since Occ3D-nuScenes lacks instance-level occupancy ground truth, we project our results onto LiDAR points for panoptic segmentation evaluation as reference metrics.

Table 9 shows a comparison of instance-level occupancy prediction for multiple methods. PanoOcc (Wang et al., 2024) remains the first purely camera-based approach for point cloud panoptic segmentation, achieving LiDAR-comparable performance through joint semantic occupancy prediction and 3D detection with bounding box supervision. However, our CSV-Occ differs fundamentally by excluding 3D bounding box size supervision (crucial for explicit instance boundary prediction).

