# OpenReview forum: "CSV-Occ: Fusing Multi-frame Alignment for Occupancy Prediction with Temporal Cross State Space Model and Central Voting Mechanism"
_ICML.cc/2025/Conference — ICML 2025 poster_

### Official Review · Reviewer_BeaF · 2025-03-05

**Overall Recommendation:** 3

**Summary:**

This paper focuses on image-based 3D semantic occupancy prediction. To address the challenges posed by the computational complexity of temporal methods and the semantic ambiguity leading to vacancy issues, the Cross-State Space Module (Cross SSM) and a Voting-based Enhancement Mechanism are proposed as targeted solutions. The approach achieves state-of-the-art performance on the OCC3D-nuScenes dataset.

**Claims And Evidence:**

This paper identifies two challenges in 3D semantic occupancy prediction:

1. **Temporal computation complexity** – While this issue does exist, the paper lacks efficiency-related experiments to substantiate this claim.
2. **Missing foreground object instance centers** – The authors demonstrate the effectiveness of their method through visualizations. However, this issue seems to stem from the fact that the ground truth data itself is based on LiDAR sequences, leading to hollow centers. Consequently, the proposed voting mechanism appears more like a post-processing optimization.

**Essential References Not Discussed:**

NA

**Experimental Designs Or Analyses:**

The experiments are conducted on occ3D-nuScenes and nuScenes, demonstrating the performance advantages of the proposed method. The ablation studies further highlight the effectiveness of the proposed modules.

**Methods And Evaluation Criteria:**

This paper proposes the Cross-State Space Module, and Table 3 demonstrates the performance advantages of the proposed fusion method. However, it does not reflect the efficiency advantages in long-term sequences.

The proposed Voting-based Enhancement Mechanism is also validated in Table 4.

**Other Comments Or Suggestions:**

No

**Other Strengths And Weaknesses:**

1. Lacks experimental comparisons on with other methods in efficiency perspective.

**Questions For Authors:**

1. The novelty of the method seems somewhat weak. What are the technical challenges of applying Mamba's SSM to the 3D occupancy prediction task?

**Relation To Broader Scientific Literature:**

Occupancy plays a crucial role in the detection of general objects. Many previous methods, such as PanoOcc and FBOcc, have explored temporal approaches. Building on these works, this paper further investigates improvements in both performance and efficiency.

**Theoretical Claims:**

No theoretical claims.

---

> ### Author Rebuttal · Authors · 2025-04-01
>
> Deeply grateful for your dedication and expertise throughout the review.
> Building on your insightful suggestions, we have systematically:
>
> 1. **Revised the paper,**
> 2. **Organized your comments by theme, and**
> 3. **Provided detailed responses in an annotated Q&A below.**
>
> We welcome any additional feedback to further refine this work.
>
> ---
>
> ### **Q1**: Temporal computation complexity – the paper lacks efficiency-related experiments to substantiate this claim it does not reflect the efficiency advantages in long-term sequences.
>
> **A1**: As shown in the following table, we have supplemented the efficiency experiments, including the number of trainable parameters, inference latency, and inference memory. The following efficiency experiments were all measured on a platform equipped with an Intel Xeon Gold 5318Y CPU and an NVIDIA A40 48G GPU, with the batch size set to 1.
>
> |Model |Fusion Module |mIoU (Occ.) |mIoU (Seg.) |Trainable Params (M) |Inference Latency (ms) |Inference Memory (MB) |
> |---|---|---|---|---|---|---|
> |CSV-Occ |Cross SSM (Ours) |**44.93** |**73.42** |**68.5** |**102** |**5588** |
> | |TSA |43.17 |72.62 |69.3 |119 |5712 |
> | |MLP-Mixer |43.22 |72.67 |77.5 |116 |6224 |
>
> We explored how initialized voxel query size impacts model efficiency (If the following images don't work, please try this link: https://github.com/ZeaZoM/re/blob/main/figs/efficiency%20experiment%20BEV.png). Keeping its height at 4, we adjusted the BEV side length from 25 to 200 (max 200×200, not exceeding occupancy ground truth). The left figure shows samples inferred per second; the right shows inference memory consumption. As voxel query size grows, especially past 100, TSA and MLP mixer's inference speed and memory consumption worsen. Cross SSM's efficiency decline is more stable due to its linear computational complexity. A larger query size means a longer flattened token sequence, and Cross SSM needs only one scan for multi frame fusion.
>
> ![](https://github.com/ZeaZoM/re/blob/main/figs/efficiency%20experiment%20BEV.png)
>
> The figure below shows how the number of inference frames affects model efficiency (If the following images don't work, please try this link: https://github.com/ZeaZoM/re/blob/main/figs/efficiency%20experiment%20Frames.png). The initialized voxel query size is set at 100×100×4. Cross SSM outperforms in both inference speed and memory consumption. Still, as the number of inference frames rises, Cross SSM's efficiency change trend is like that of TSA and MLP mixer. In CSV - Occ, the number of inference frames equals the number of multi - frame fusion module calls, and the feature sequence length per call depends only on the voxel query size. So, more inference frames don't give Cross SSM a trend advantage.
>
> ![](https://github.com/ZeaZoM/re/blob/main/figs/efficiency%20experiment%20Frames.png)
>
> ### **Q2**: The novelty of the method seems somewhat weak. What are the technical challenges of applying Mamba's SSM to the 3D occupancy prediction task?
>
> **A2**: Our core challenge lies in extending Mamba's SSM to process dual-sequence inputs - a novel direction beyond existing vision-focused Mamba variants that primarily optimize feature map scanning for single-sequence SSM receptive fields. While standard "self SSM" (analogous to self-attention) updates features through intra-sequence correlations, we pioneer "cross SSM" to enable inter-sequence interaction akin to cross-attention.
>
> The validation complexity stems from SSM's six parametric elements (x, A, B, C, D, Δ). Unlike attention where k/v derive from one sequence and q from another, SSM's A/D originate from learnable embeddings while B/C/Δ are mapped from the input sequence. Our key hurdle was determining which sequences should supply x/B/C/Δ for cross-SSM implementation, as the original SSM formulation provides no guidance. Through systematic permutation experiments, we empirically identified optimal configurations.
>
> |# |x |C |B |Δ |mIoU (Occ.) |mIoU (Seg.) |
> |---|---|---|---|---|---|---|
> |A |$V_{T-i}$ |$V_{T}^{'}$ |$V_{T-i}$ |$V_{T}^{'}$ |39.35 |67.19 |
> |B | | |$V_{T}^{'}$ |$V_{T-i}$ |38.60 |67.01 |
> |C | | |$V_{T-i}$ |$V_{T-i}$ |**44.93** |**73.42** |
> |D |$V_{T}^{'}$ |$V_{T-i}$ |$V_{T-i}$ |$V_{T}^{'}$ |33.25 |61.07 |
> |E | | |$V_{T}^{'}$ |$V_{T-i}$ |35.84 |64.50 |
> |F | | |$V_{T}^{'}$ |$V_{T}^{'}$ |32.53 |59.28 |
>
> Empirical analysis revealed Approach #C's superior metrics prompted theoretical investigation into SSM-attention parallels. Through mathematical derivation (Appendix B), we demonstrated SSM constitutes a specialized attention subset, cross-attention's Q→C substitution implies cross-SSM implementation requires mapping C from secondary sequences - a pivotal insight enabling cross-sequence feature interaction.
>
> Methodologically, we preserved original SSM's feature-map scanning architecture (patch partition via pooling/flattening; merger via reshape/interpolation - cf. main text line 199) to isolate cross-SSM effects.

---

> > ### Comment · Reviewer_BeaF · 2025-04-05
> >
> > The authors have provided additional experiments on efficiency. While the improvements are not particularly significant, they are still meaningful and demonstrate the potential value of the method. It would be interesting to further evaluate its effectiveness under higher-resolution settings, such as on Occ-Waymo. Based on these revisions, I am raising my score to a weak accept.

---

### Official Review · Reviewer_QyW7 · 2025-03-06

**Overall Recommendation:** 3

**Summary:**

The paper introduces CSV-Occ, a method for camera-based 3D semantic occupancy prediction.
CSV-Occ focuses on two key challenges.

Firstly, the prior methods have usually exploited attention mechanisms for temporal modeling that have high computational complexity. This paper propose the Cross State Space Module (Cross SSM), a variant of Mamba architecture, to handle multi-sequence inputs with linear complexity.

In addition, the existing methods struggle with Internal Occupancy Vacancy (IOV), where the centers of large foreground objects are often predicted as empty. To address this, the authors propose the Voting-based Enhancement Mechanism (VEM), which refines semantic occupancy predictions by inferring the instance centers of objects.

SCV-Occ achieves SoTA performance on Occ3D-nuScenes dataset.

**Claims And Evidence:**

In my opinion, the claims in the paper are generally well-supported.
Experimental results on the Occ3D-nuScenes dataset confirm that CSV-Occ outperforms existing methods on 3D semantic occupancy prediction and LiDAR semantic segmentation.

**Essential References Not Discussed:**

It would be much better if the authors refer more recent works, such as OccFiner [1] or TALoS [2].

[1] OccFiner: Offboard Occupancy Refinement with Hybrid Propagation for Autonomous Driving

[2] TALoS: Enhancing Semantic Scene Completion via Test-time Adaptation on the Line of Sight

**Experimental Designs Or Analyses:**

The experimental designs are solid. The paper uses well-established datasets for evaluation, and the ablation studies are extensive enough to explore the impact of different components like the Cross SSM and VEM.
But more extensive validation might be helpful (more dataset including indoor cases).

**Methods And Evaluation Criteria:**

The proposed methods are sounding for the defined problems.
The Cross SSM is a nice solution for efficient temporal feature fusion, which is crucial for occupancy prediction in autonomous driving.
The VEM is slightly less novel in terms of academia, but still a meaningful technical solution for improving occupancy predictions, particularly for large objects.
And also the evaluation criterial seems appropriate.
The authors evaluate CSV-Occ against existing baselines using the Occ3D-nuScenes dataset and achieve good results.

**Other Comments Or Suggestions:**

Please see the reviews above.

**Other Strengths And Weaknesses:**

The approach combines cutting-edge methods (Mamba, especially) for temporal feature fusion and semantic occupancy prediction.
The empirical results demonstrate clear improvements over previous methods.

From my perspective, the academic novelty might be somewhat lacking, but the contribution to the field is substantial.

**Questions For Authors:**

Have you explored using indoor point cloud data in conjunction with your method?
The targeted problems are also important issues in that field.

**Relation To Broader Scientific Literature:**

Literature review of this paper is pretty good, particularly in the area of 3D semantic occupancy prediction and temporal feature fusion.

**Theoretical Claims:**

It seems that there is no special theoretical claims that needs validation.
The proposed method is verified with experimental results.

---

> ### Author Rebuttal · Authors · 2025-04-01
>
> We sincerely express our profound gratitude for the precious time and dedicated effort you have invested in the review process. Taking into account the highly constructive feedback and suggestions you proffered, we have meticulously re-examined our paper and work. **We have comprehensively collated your comments and will respond to each of them individually in a Question-and-Answer format as detailed below.** Should you have any further questions or concerns, we are wholeheartedly committed to collaborating with you to resolve them.
>
> ---
>
> ### **Q1**: Have you explored using indoor point cloud data in conjunction with your method? The targeted problems are also important issues in that field.
>
> **A1**: CSV-Occ focuses on the sub-task of pure image-based semantic occupancy prediction in outdoor autonomous driving. Thus, we haven't explored applying it to indoor scene datasets or integrating indoor point cloud data for validation. However, in future work, we will strongly consider your suggestion to transfer our method to indoor scene datasets and extend it to multi-modal inputs that combine images and point clouds, in order to evaluate the generality of CSV-Occ.
>
> ### **Q2**: It would be much better if the authors refer more recent works, such as OccFiner or TALoS.
>
> **A2**: We fully agree.
> 1. OccFiner focuses on achieving a data closed loop and automatic annotation for pure vision SSC. In the first stage, it compensates for the prediction errors of the in-vehicle model through a multi-to-multi local propagation network, and fuses the relative spatial coordinates and semantic features. In the second stage, it conducts global propagation of the regional centers, converts the refined voxel labels into semantic point clouds, adjusts the coordinates, and conducts voxel voting. **We will cite it in our article: 'Shi H, Wang S, Zhang J, et al. Offboard Occupancy Refinement with Hybrid Propagation for Autonomous Driving[J]. arXiv preprint arXiv:2403.08504, 2024.'**
>
> 2. TALoS mines the information in the driving environment, uses the point cloud observations at different moments as supervision. Through coordinate transformation, it obtains binary self-supervision for geometric completion based on the characteristics of LiDAR's line of sight, and constructs a loss function to guide the model training. **We will also cite it in our article: 'Jang H K, Kim J, Kweon H, et al. TALoS: Enhancing Semantic Scene Completion via Test-time Adaptation on the Line of Sight[J]. arXiv preprint arXiv:2410.15674, 2024.'**
>
> 3. The key insight of CVT-Occ is that as the camera moves with the vehicle, parallax is generated. By leveraging this multi-frame parallax information, the depth information that cannot be directly obtained from a single pixel can be compensated for, thus eliminating the uncertainty that occurs when converting image features into the 3D space. Methodologically, CVT-Occ projects rays from the center of the current-frame BEV feature volume towards the image pixels. Then, it samples multiple virtual points and their corresponding voxel features at a certain step along the rays. These virtual points are transformed into the historical BEV feature volume through coordinate system transformation to sample the corresponding voxel features. All the sampled voxel features are combined to form Cost Volume Features. After convolution and Sigmoid calculation, attention weights are generated, which are multiplied voxel-by-voxel with the current-frame BEV feature volume to update the features. **We will discuss and cite it in our paper: 'Ye Z, Jiang T, Xu C, et al. Cvt-occ: Cost volume temporal fusion for 3d occupancy prediction[C]//European Conference on Computer Vision. Cham: Springer Nature Switzerland, 2024: 381-397.'**

---

### Official Review · Reviewer_9BHX · 2025-03-15

**Overall Recommendation:** 3

**Summary:**

The paper proposes CSV-Occ, a method for image-based 3D semantic occupancy prediction in autonomous driving. It introduces two key ideas: 1. Temporal fusion applied on voxel query results instead of BEV features, which is considered new. 2. A center voting mechanism to improve occupancy prediction inside object boundaries. Despite these contributions, the paper lacks an efficiency analysis to justify the use of state-space modeling (SSM) for computational benefits. Additionally, while the voting feature is fused into the network, no panoptic results are provided.

**Claims And Evidence:**

* No direct evaluation or quantitative analysis is provided to compare efficiency with self-attention-based methods. While reducing complexity from quadratic to linear is theoretically appealing, empirical proof (e.g., runtime comparison, FLOPs, inference speed) is missing.
* The claim is only supported by Figure 7, but lacks statistical analylsis to reveal the mechanism.

**Essential References Not Discussed:**

"CVT-Occ: Cost Volume Temporal Fusion for 3D Occupancy Prediction" (ECCV 2024), which is highly relevant and should be discussed.

**Experimental Designs Or Analyses:**

+ Ablation studies are provided at an architectural level, following standard experimental protocols. The model is evaluated on the Occ3D-nuScenes dataset, which is an appropriate benchmark.
- The projection of vision-only 3D semantic occupancy predictions onto point clouds for evaluation against LiDAR semantic segmentation is questionable.

**Methods And Evaluation Criteria:**

* The paper introduces center voting but does not provide panoptic or instance understanding results, which would be the natural evaluation metric. Evaluating thing-stuff separation and instance-wise completeness would better support the effectiveness of the voting feature.
* Temporal fusion can be applied beyond occupancy prediction, e.g., mapping, object detection and end-to-end planning. A broader evaluation would show whether CSV is specific to occupancy prediction or applicable to other self-driving tasks.

**Other Comments Or Suggestions:**

- Time-based view transformation is essentially just view transformation—terminology clarification is needed.

**Other Strengths And Weaknesses:**

- No SemanticKITTI experiments, which would help assess generalization beyond nuScenes.
- Figure 2 caption might be incorrect?

**Questions For Authors:**

* Since center voting is already implemented, why not directly generate instance-level occupancy predictions?

**Relation To Broader Scientific Literature:**

* The paper could have cited more occupancy prediction literature, including indoor scene methods, which share methodological similarities.

**Theoretical Claims:**

* The mathematical formulation for SSM-based temporal fusion is included.

---

> ### Author Rebuttal · Authors · 2025-04-01
>
> We sincerely appreciate your time and effort in the review process. **In response to your constructive feedback, we have thoroughly reviewed our paper and categorized your comments. Attached is Q&A response.** If you have any further questions, we will address them promptly.
>
> ---
>
> ### **Q1**: The projection of vision-only 3D semantic occupancy predictions onto point clouds for evaluation against LiDAR semantic segmentation is questionable.
>
> **A1**: When the volume of a single voxel shrinks to near zero, the semantic occupancy prediction task can approximate the point cloud semantic segmentation task. Semantic occupancy is a voxelized 3D point representation. Our evaluation method, using LiDAR semantic segmentation as a quantitative indicator for 3D semantic occupancy prediction, follows these relevant works: PanoOcc (CVPR'24), TPVFormer (CVPR'23), OccFormer (ICCV'23), Scene as Occupancy (ICCV'23)
>
> ### **Q2**: Since center voting is already implemented, why not directly generate instance-level occupancy predictions?
>
> **A2**: As you said, we can generate instance-level results, and CSV-Occ requires additional post-processing. Our implementation involves three key steps:
> 1. Applying Relative Central Regression to the final semantic occupancy output for voxel-level center prediction
> 2. Clustering potentially scattered center predictions (Due to the error in the center voting predicted by the model, multiple discrete center points often appear) to form coherent instances
> 3. The clustered central voxels then determine instance ID assignments through voting relationships. Since Occ3D-nuScenes lacks instance-level occupancy ground truth, we project our results onto LiDAR points for panoptic segmentation evaluation as reference metrics.
>
> |Method |Modality |PQ |SQ |RQ |
> |---|---|---|---|---|
> |PanopticTrackNet (arXiv 2020) |L |51.4 |80.2 |63.3 |
> |EfficientLPS (I-TR 2021) |L |62.0 |83.4 |73.9 |
> |LidarMulitiNet (AAAI 2023) |L |81.8 |89.7 |90.8 |
> |PanoOcc (CVPR 2024) |C |62.1 |82.1 |75.1 |
> |CSV-Occ-Instance (Ours) |C |48.3 |79.6 |60.5 |
>
> PanoOcc remains the first purely camera-based approach for point cloud panoptic segmentation, achieving LiDAR-comparable performance through joint semantic occupancy prediction and 3D detection with bounding box supervision. However, our CSV-Occ differs fundamentally by excluding 3D bounding box size supervision (crucial for explicit instance boundary prediction). This leads to center-voting derived instance results underperforming 2020 LiDAR-based PanopticTrackNet in evaluations. If you think it is necessary, we can consider adding the above results to the appendix of the article for analysis.
>
> ### **Q3**: The claim of VEM is only supported by Figure 7, but lacks statistical analysis to reveal the mechanism.
>
> **A3**: Regarding the support for VEM, apart from Figure 7, we also did a quantitative ablation test in Table 4. Yet, as you suggested, a more detailed category-specific statistical analysis of VEM is needed. We found that the VEM module can cause a decrease in the performance of certain background categories. **For detailed charts and analysis (constrained by 5k-character rebuttal limits), please see Section A2 (Reviewer a93Q). Hope it eases your concerns.**
>
> ### **Q4**: No direct evaluation or quantitative analysis is provided to compare efficiency with self-attention-based methods.
>
> **A4**: Per your feedback, we conducted efficiency analyses on Cross SSM, evaluating voxel query size and frame count impacts. Results show that Cross SSM has the optimal trainable parameters, inference speed, and memory. **For detail, please see Section A1 (Reviewer BeaF).**
>
> ### **Q5**: CVT-Occ (ECCV'24), which is highly relevant and should be discussed.
>
> **A5**: We will cite it. **For our specific discussion on CVT-Occ, please refer to A2 (Reviewer QyW7).**
>
> ### **Q6**: No SemanticKITTI experiments.
>
> **A6**: Since nuScenes has a larger sample size and each sample consists of six camera images forming a 360 degree surround view field of view. We evaluated semantic occupancy on the Occ3D-nuScenes dataset and point cloud semantic segmentation on the nuScenes dataset. In recent related works, there are also many that only evaluate on the nuScenes dataset, such as OPUS (NeurIPS'24), COTR (CVPR'24), Fully Sparse (ECCV'24), FB-OCC (ICCV'23).
>
> Therefore, we believe two experiments suffice to comprehensively demonstrate the effectiveness of our method. **We will still seriously consider incorporating SemanticKITTI in our future work.**
>
> ### **Q7**: Time-based view transformation-terminology clarification is needed.
>
> **A7**: To resolve ambiguity in Section 3.3’s original title "Time-based View Transformation":
> 1. Retitle to "Time-based Feature Fusion" (TFF)
> 2. Add Subsection 3.3.1 "View Transformation" to clarify content scope
> 3. Renumber subsequent subsections (3.3.1→3.3.2) and update Figure 3’s "TVT" labels to "TFF"
> 4. Systematically replace all "TVT" abbreviations with "TFF" throughout the paper

---

> > ### Comment · Reviewer_9BHX · 2025-04-04
> >
> > An ICML-level paper should tackle panoptic occupancy prediction if center voting is the key technique. Additionally, the provided statistical analysis and efficiency evaluations feel marginal and do not sufficiently support the claimed advantages. The absence of SemanticKITTI experiments also limits the generalizability claims of the method.
> >
> > Furthermore, I find the term "Time-based Feature Fusion" to be unnecessarily vague. The distinction between view transformation and feature fusion is well-understood in the literature, and it's unclear what exactly is meant by “time-based” in this context. My comment 'Time-based view transformation is essentially just view transformation' remains valid.
> >
> > I've updated my recommendation to a WA. No further response is needed.

---

### Official Review · Reviewer_a93Q · 2025-03-16

**Overall Recommendation:** 4

**Summary:**

This paper presents CSV-Occ, a method for camera-based 3D semantic occupancy prediction, aimed at improving scene understanding. It considers two key issues: reducing the high computational complexity of temporal information fusion and addressing the semantic ambiguity in predicting object centers. To overcome these challenges, CSV-Occ extends the state space model to support multi-input sequence interactions and explicitly predicts the instance to which each voxel belongs, refining feature representation from coarse to fine. Experiments on the Occ3D-nuScenes and nuScenes datasets demonstrate that CSV-Occ performs better than some of the existing methods in both 3D semantic occupancy prediction and lidar point cloud semantic segmentation.

## update after rebuttal
Read through the authors response and I'll keep my current rating.

**Claims And Evidence:**

The claims are generally well-supported by thorough experimentation, with near-perfect results. However, for some classes, the results still do not reach state-of-the-art (SOTA) levels, and the lack of explanation for these cases is a noticeable gap. Nevertheless, the claims are largely substantiated through the use of ablation studies, which provide strong support for the findings.

**Essential References Not Discussed:**

I am not aware of any essential references that were not covered in the paper.

**Experimental Designs Or Analyses:**

The experimental designs are well-structured, with a thorough comparison to multiple approaches. However, the proposed method does not perform as well on certain classes, such as trailers, vegetation, and manmade objects. While this is understandable given the focus on foreground classes, it would be beneficial if the authors provided some discussion on why the performance on background classes is slightly worse compared to state-of-the-art (SOTA) methods.

**Methods And Evaluation Criteria:**

The evaluation criteria are well-founded, and the authors compare their proposed method with several other state-of-the-art (SOTA) approaches. To ensure a fair comparison, they even modified the FB-OCC code. The metric used for evaluation is mean Intersection over Union (mIoU), which is also reported for each class label.

**Other Comments Or Suggestions:**

NA

**Other Strengths And Weaknesses:**

As mentioned above, providing an explanation for the lack of performance on certain classes would add more value to the paper. Additionally, a qualitative analysis comparing the proposed approach with SOTA methods, highlighting where it excels and where it falls short, would also be helpful for better understanding its strengths and limitations.

**Questions For Authors:**

1. Could you elaborate on the reasons for the lower performance on certain background classes, such as trailers and vegetation?

2. Was there any consideration of integrating additional data sources, such as lidar or radar, to enhance the accuracy of scene understanding?

**Relation To Broader Scientific Literature:**

The proposed approach is highly relevant to the field of scene understanding, particularly in the context of autonomous driving. By improving 3D semantic occupancy prediction, it provides valuable insights into how vehicles can better perceive and interpret their surroundings. Furthermore, the approach contributes to the broader literature by offering a more efficient and effective way to handle complex environmental scenarios, which has potential applications beyond just autonomous driving, such as robotics and urban planning.

**Theoretical Claims:**

The theoretical claims presented in the main paper appear promising. However, a significant portion of the theoretical details is covered in the supplementary section, such as the "Derivations of the Cross State Space Module" and evaluation metric formulas. As a result, the authors rely on the supplementary material to fully convey the theoretical claims.

---

> ### Author Rebuttal · Authors · 2025-04-01
>
> We sincerely appreciate the valuable time and effort you've dedicated during the review process. In light of the constructive feedback and suggestions you provided, we've meticulously examined our paper and work. **We've also summarized your comments and are replying to you one by one in a Question and Answer format as follows.** If you have any further questions or concerns, we're more than willing to cooperate with you to address them.
>
> ---
>
> ### **Q1**: Was there any consideration of integrating additional data sources, such as lidar or radar, to enhance the accuracy of scene understanding?
>
> **A1**: We haven't integrated multiple sensors (such as LiDAR or radar) into CSV-Occ yet. In the future, we'll seriously consider your suggestion of incorporating multi-sensor multi-modal inputs into our method and further applying it to more scenarios (like indoor scene datasets or more vision tasks) to evaluate its generality.
>
> ### **Q2**: Could you elaborate on the reasons for the lower performance on certain background classes, such as trailers and vegetation?
>
> **A2**: Through statistics, we've found that the low performance of our method in certain background categories is primarily attributed to the VEM module. As depicted in the following figure, upon enabling the voting mechanism, the performance of all foreground categories has increased. Conversely, a decline in performance is observed in four background categories: driveable surface, other flat, man-made, and vegetation. (If the following images don't work, try this link: https://github.com/ZeaZoM/re/blob/main/figs/VEM%20cat.png)
>
> ![](https://github.com/ZeaZoM/re/blob/main/figs/VEM%20cat.png)
>
> This occurs because VEM relies on the coarse semantic occupancy prediction to divide foreground category voxels. When the coarse semantic occupancy prediction is inaccurate, VEM may predict relative centers for some background voxels as well. The occupancy of background categories usually has a large volume and lacks distinct instance centers. This leads to training confusion in VEM. During inference, it wrongly updates the features of background voxels, causing the occupancy classification head to give incorrect category predictions. As a result, the number of background voxels decreases, which directly impacts the prediction performance of background categories.
>
> We also calculated the proportion of the number of foreground and background voxels predicted by the model with and without VEM. As shown in the figure below, it can be seen that VEM can significantly increase the number of foreground voxels. This partly indicates that VEM can improve the Internal Occupancy Vacancy (IOV) situation and successfully predict free voxels as foreground voxels. However, we unexpectedly found that the number of background voxels decreased, which we think is related to the training confusion caused by VEM. (If the following images don't work, please try this link: https://github.com/ZeaZoM/re/blob/main/figs/VEM%20ratio.png)
>
> ![](https://github.com/ZeaZoM/re/blob/main/figs/VEM%20ratio.png)
>
> ### **Q3**: The section that qualitatively evaluates the proposed approach against other SOTA methods is very helpful. It would be beneficial to move this content to the main paper, as qualitative analysis provides more insight into the impact of the approach than numbers alone.
>
> **A3**: Since the page limit for the main text when submitting the article is no more than 8 pages, we previously had to move some content to the appendix. However, in line with your suggestion, we will consider moving the qualitative analysis content from the appendix to the main text within the scope permitted by the article layout.

---

> > ### Comment · Reviewer_a93Q · 2025-04-03
> >
> > Read through the authors response and I'll keep my current rating.

---

### Decision · Program_Chairs · 2025-05-01

**Decision:**

Accept (poster)

**Comment:**

The paper initially received ratings of (4, 2, 3, 2), which were somewhat varied. After the rebuttal, the ratings increased to (4, 3, 3, 3).

Although the final ratings are positive, reviewers raised concerns regarding the statistical analysis and efficiency evaluations, which do not adequately support the claimed advantages. The lack of SemanticKITTI experiments also limits the generalizability claims of the method. Reviewers also noted that the overall contribution of the paper seems somewhat modest, leaving them uncertain about its suitability for inclusion at ICML. Further revisions are suggested to clarify terminology related to "Time-based Feature Fusion" and "Time-based View Transformation," and to evaluate the method's effectiveness under higher-resolution settings, such as on Occ-Waymo.

Despite the concerns, the area chair found no clear justification to overrule the reviewers' recommendations. Consequently, the area chair suggests a "weak accept" if there is room in the program